# Noise or Signal: The Role of Image Backgrounds in Object Recognition

**Kai Xiao, Logan Engstrom, Andrew Ilyas, Aleksander Mądry**
MIT
`{kaix,engstrom,ailyas,madry}@mit.edu`

## Abstract

We assess the tendency of state-of-the-art object recognition models to depend on signals from image backgrounds. We create a toolkit for disentangling foreground and background signal on ImageNet images, and find that (a) models can achieve non-trivial accuracy by relying on the background alone, (b) models often misclassify images even in the presence of correctly classified foregrounds—up to 88% of the time with adversarially chosen backgrounds, and (c) more accurate models tend to depend on backgrounds less. Our analysis of backgrounds brings us closer to understanding which correlations machine learning models use, and how they determine models' out of distribution performance.

## 1 Introduction

Object recognition models are typically trained to minimize loss on a given dataset, and evaluated by the accuracy they attain on the corresponding test set. In this paradigm, model performance can be improved by incorporating any generalizing correlation between images and their labels into decision-making. However, the actual model reliability and robustness depend on the specific set of correlations that is used, and on how those correlations are combined. Indeed, outside of the training distribution, model predictions can deviate wildly from human expectations either due to relying on correlations that humans do not perceive (Jetley et al., 2018; Ilyas et al., 2019; Jacobsen et al., 2019), or due to overusing correlations, such as texture (Geirhos et al., 2019; Baker et al., 2018) and color (Yip & Sinha, 2002), that humans do use (but to a lesser degree). Characterizing the correlations that models depend on thus has important implications for understanding model behavior, in general.

Image backgrounds are a natural source of correlation between images and their labels in object recognition. Indeed, prior work has shown that models may use backgrounds in classification (Zhang et al., 2007; Ribeiro et al., 2016; Zhu et al., 2017; Rosenfeld et al., 2018; Zech et al., 2018; Barbu et al., 2019; Shetty et al., 2019; Sagawa et al., 2020; Geirhos et al., 2020), and suggests that even human vision makes use of image context for scene and object recognition (Torralba, 2003). In this work, we aim to obtain a deeper and more holistic understanding of how current state-of-the-art image classifiers utilize image backgrounds. To this end, in contrast to most of the prior work (which tends to study relatively small and often newly-curated image datasets[1]), our focus is on ImageNet (Russakovsky et al., 2015)—one of the largest and most widely used datasets, with state-of-the-art training methods, architectures, and pre-trained models tuned to work well for it.

Zhu et al. (2017) analyze ImageNet classification (focusing on the older, AlexNet model) to find that AlexNet achieves small but non-trivial test accuracy on a dataset consisting of only backgrounds (where foreground objects are replaced by black rectangles). While sufficient for establishing that backgrounds can be used for classification, we aim to go beyond those initial explorations to get a more fine-grained understanding of the relative importance of backgrounds and foregrounds, for newer, state-of-the-art models, and to provide a versatile toolkit for others to use. Specifically, we investigate the extent to which models rely on backgrounds, the implications of this reliance, and how models' use of backgrounds has evolved over time. Concretely:

- We create a suite of datasets that help disentangle (and control for different aspects of) the impact of foreground and background signals on classification. The code and datasets

---

[1]We discuss these works in greater detail in Section 5, Related Works.

are publicly available for others to use in this repository: `https://github.com/MadryLab/backgrounds_challenge`.

- Using the aforementioned toolkit, we characterize models' reliance on image backgrounds. We find that image backgrounds alone suffice for fairly successful classification and that changing background signals decreases average-case performance. In fact, we further show that by choosing backgrounds in an adversarial manner, we can make standard models misclassify 88% of images as the background class.

- We demonstrate that standard models not only use but *require* backgrounds for correctly classifying large portions of test sets (35% on our benchmark).

- We study the impact of backgrounds on classification for a variety of classifiers, and find that models with higher ImageNet test accuracy tend to simultaneously have higher accuracy on image backgrounds alone and have greater robustness to changes in image background.

## 2 METHODOLOGY

To properly gauge image backgrounds' role in image classification, we construct a synthetic dataset for disentangling background from foreground signal: ImageNet-9.

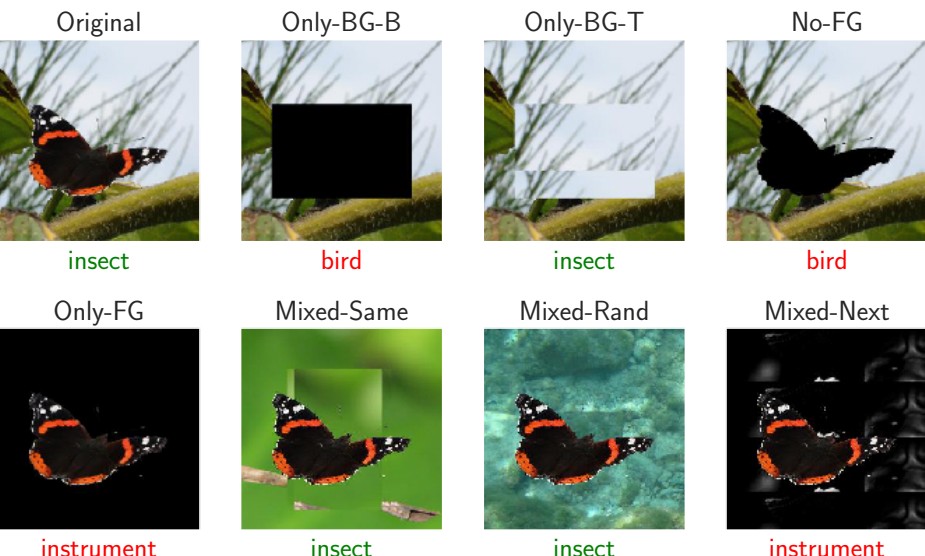

Figure 1: Variations of the synthetic dataset ImageNet-9, as described in Table 1. We label each image with its pre-trained ResNet-50 classification—green, if corresponding with the original label; red, if not. The model correctly classifies the image as "insect" when given: the original image, only the background, and two cases where the original foreground is present but the background changes. Note that, in particular, the model fails in two cases when the original foreground is present but the background changes (as in MIXED-NEXT or ONLY-FG).

**Base dataset: ImageNet-9.** We organize a subset of ImageNet into a new dataset with nine coarse-grained classes and call it ImageNet-9 (IN-9) [2]. To create it, we group together ImageNet classes sharing an ancestor in the WordNet (Miller, 1995) hierarchy. We use coarse-grained classes because there are not enough images with annotated bounding boxes (which we need to disentangle backgrounds and foregrounds) to use the standard labels. The resulting IN-9 dataset is class-balanced and has 45405 training images and 4050 testing images. While we can (and do) apply our methods on the full ImageNet dataset as well, we choose to focus on this coarse-grained version of ImageNet because of its higher-fidelity images. We describe the dataset creation process in detail and discuss the advantages of focusing on IN-9 in Appendix A.

---

[2] These classes are `dog`, `bird`, `vehicle`, `reptile`, `carnivore`, `insect`, `instrument`, `primate`, and `fish`.

**Variations of ImageNet-9** From this base set of images, which we call the ORIGINAL version of IN-9, we create seven other synthetic variations designed to understand the impact of backgrounds. We use both rectangular bounding boxes and the foreground segmentation algorithm GrabCut (Rother et al., 2004), as implemented in OpenCV, to disentangle backgrounds and foregrounds. We visualize these variations in Figure 1, and provide a detailed reference in Table 1. These subdatasets of IN-9 differ only in how they process the foregrounds and backgrounds of each constituent image.

**Larger dataset: IN-9L** We finally create a dataset called IN-9L that consists of all the images in ImageNet corresponding to the classes in ORIGINAL (rather than just the images that have associated bounding boxes). This dataset has about 180k training images in total. We leverage this larger dataset to train better generalizing models, and prefer to analyze models trained on IN-9L whenever possible.

Table 1: The 8 modified subdatasets created from ImageNet-9, which are visualized in Figure 1. The foreground detection method refers to how the pixels corresponding to the foreground are found. GrabCut refers to the foreground segmentation algorithm implemented in OpenCV. Random backgrounds in the last three datasets are taken from ONLY-BG-T. For more details see Appendix A.

| Name | Foreground | Background | Foreground Detection Method |
|---|---|---|---|
| ORIGINAL | Unmodified | Unmodified | — |
| ONLY-BG-B | Black | Unmodified | Bounding Box |
| ONLY-BG-T | Tiled background | Unmodified | Bounding Box |
| NO-FG | Black | Unmodified | GrabCut |
| ONLY-FG | Unmodified | Black | GrabCut |
| MIXED-SAME | Unmodified | Random BG of the same class | GrabCut |
| MIXED-RAND | Unmodified | Random BG of a random class | GrabCut |
| MIXED-NEXT | Unmodified | Random BG of the next class | GrabCut |

## 3 QUANTIFYING RELIANCE ON BACKGROUND SIGNALS

With ImageNet-9 in hand, we now assess the role of image backgrounds in classification.

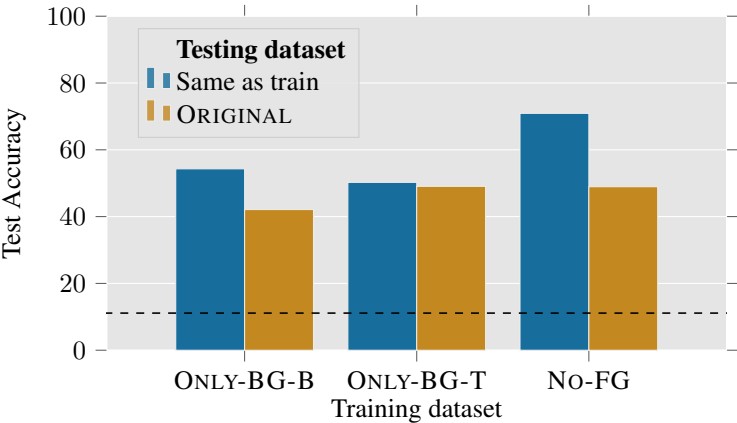

Figure 2: We train models on each of the "background-only" datasets, then evaluate each on its corresponding test set as well as the ORIGINAL test set. Even though the model only learns from background signal, it achieves (much) better than random performance on *both* the corresponding test set and ORIGINAL. Here, random guessing would give 11.11% (the dotted line).

**Backgrounds suffice for classification.** Prior work has found that models are able to make accurate predictions based on backgrounds alone; we begin by directly quantifying this ability. Looking at the ONLY-BG-T, ONLY-BG-B, and NO-FG datasets, we find (cf. Figure 2) that models trained on these background-only training sets generalize reasonably well to both their corresponding test sets and to unmodified images from the ORIGINAL test set (around 40-50% for every model, far above the

Table 2: Performance of state-of-the-art computer vision models on selected test sets of ImageNet-9. We include both pre-trained ImageNet models and models of different architectures that we train on IN-9L. The BG-GAP is defined as the difference in test accuracy between MIXED-SAME and MIXED-RAND and helps assess the tendency of such models to rely on background signal. Architectures are sorted by their test accuracies on ImageNet and ORIGINAL for pre-trained and IN-9L-trained models, respectively. Shaded in grey are the two architectures that can be directly compared across datasets (ResNet-50 and Wide-ResNet-50x2).

| | Pre-trained on ImageNet | | | | | Trained on IN-9L | | | | |
|---|---|---|---|---|---|---|---|---|---|---|
| Test dataset | MobileNet-v3 | EfficientNet-b0 | ResNet-50 | WRN-50x2 | DPN-92 | AlexNet | ShuffleNet | ResNet-50 | WRN-50x2 | VGG16-BN |
| ImageNet | 67.9% | 77.2% | 77.6% | 78.5% | 80.0% | | | —— | | |
| ORIGINAL | 95.5% | 96.1% | 96.9% | 96.6% | 97.2% | 86.7% | 95.7% | 96.3% | 97.2% | 97.6% |
| ONLY-BG-T | 16.3% | 16.5% | 17.4% | 18.8% | 17.6% | 41.5% | 43.6% | 43.6% | 45.1% | 45.7% |
| MIXED-SAME | 84.0% | 86.2% | 91.0% | 88.3% | 90.5% | 76.2% | 86.7% | 89.9% | 90.6% | 91.0% |
| MIXED-RAND | 73.2% | 76.3% | 84.3% | 81.4% | 86.1% | 54.2% | 69.4% | 75.6% | 78.0% | 78.0% |
| BG-gap | 10.8% | 9.9% | 6.7% | 6.9% | 4.4% | 22.0% | 17.3% | 14.3% | 12.6% | 13.0% |

baseline of 11% representing random classification). Our results confirm that image backgrounds contain signal that models can accurately classify standard images with.

**Models exploit background signal for classification.**    We discover that models can misclassify due to background signal, especially when the background class does not match that of the foreground. As a demonstration, we study model accuracies on the MIXED-RAND dataset, where image backgrounds are randomized and thus provide no information about the correct label. By comparing test accuracies on MIXED-RAND and MIXED-SAME [3], where images have class-consistent backgrounds, we can measure classifiers' dependence on the correct background. We denote the resulting accuracy gap between MIXED-SAME and MIXED-RAND as the BG-GAP; this difference represents the drop in model accuracy due to changing the class signal from the background. In Table 2, we observe a BG-GAP of 13-22% and 4-11% for models trained on IN-9L and ImageNet, respectively, suggesting that backgrounds often mislead state-of-the-art models *even when the correct foreground is present*.

**More Training Data can reduce the BG-GAP.** Our results indicate that ImageNet-trained models are less dependent on backgrounds than their IN-9L-trained counterparts—they have a smaller (but still significant) BG-GAP, and perform worse when predicting solely based on backgrounds (i.e., on the ONLY-BG-T dataset). We explore two ways that ImageNet differs from IN-9L to understand this phenomena—ImageNet has (a) more datapoints than IN-9L, and (b) a more fine-grained class structure. Figure 3 shows that more training data reduces the BG-GAP, particularly when the training dataset size approaches the size of ImageNet. This indicates that training on much more data (and thus, more backgrounds) can reduce (but not eliminate) the effect of backgrounds on model predictions. An ablation study of ImageNet's more fine-grained class structure does not find strong evidence supporting its helpfulness (cf. Appendix B).

**Models are vulnerable to adversarial backgrounds.**    To understand how worst-case backgrounds impact models' performance, we evaluate model robustness to adversarially chosen backgrounds. We find that 88% of foregrounds are susceptible to such backgrounds; that is, for these foregrounds, there is a background that causes the classifier to classify the resulting foreground-background combination as the background class. For a finer grained look, we also evaluate image backgrounds based on their attack success rate (ASR), i.e., how frequently they cause models to predict the (background) class in the presence of a conflicting foreground class. As an example, Figure 4 shows the five backgrounds with the highest ASR for the insect class—these backgrounds (extracted from insect images in ORIGINAL) fool a IN-9L-trained ResNet-50 model into predicting insect on up to 52% of

---

[3]MIXED-SAME controls for artifacts from image processing presented in MIXED-RAND. For further discussion, see Appendix D.

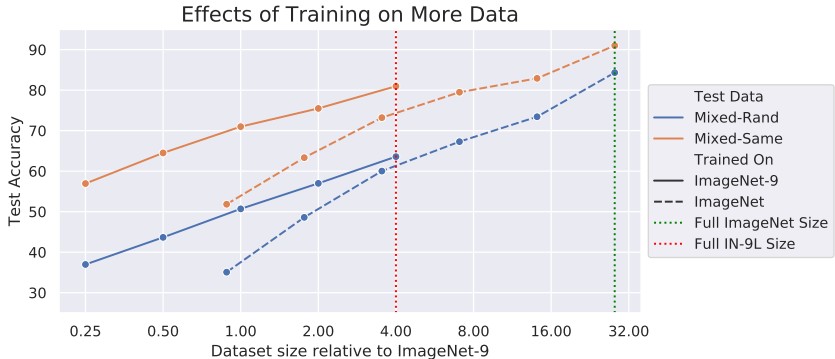

Figure 3: We compare test accuracies on MIXED-SAME and MIXED-RAND and observe that training with more data reduces the BG-GAP (BG-GAP measures the effect of backgrounds on model predictions). While this trend is true for models trained on both IN-9 and ImageNet, the trend is most noticeable for models trained on the largest training set, the full ImageNet dataset—this is shown on the far right side of the graph.

non-insect foregrounds. We plot a histogram of ASR over all insect backgrounds in Figure 24 of the Appendix—it has a long tail. Similar results are observed for other classes as well (cf. Appendix G).

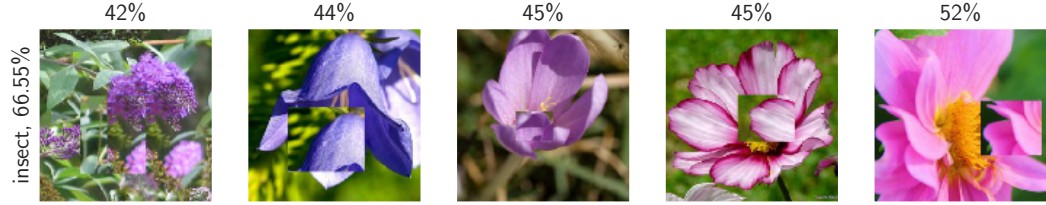

Figure 4: The adversarial backgrounds that most frequently fool IN-9L-trained models into classifying a given foreground as insect, ordered by the percentage of foregrounds fooled. The total portion of images that can be fooled (by any background from this class) is 66.55%.

**Training on MIXED-RAND reduces background dependence.** Next, we explore how to reduce models' dependence on background. To this end, we train models on MIXED-RAND, a synthetic dataset where background signals are decorrelated from class labels. As we would expect, MIXED-RAND-trained models extract less signal from backgrounds: evaluation results show that MIXED-RAND models perform poorly (15% accuracy—barely higher than random) on datasets with only backgrounds and no foregrounds, (ONLY-BG-T or ONLY-BG-B).

Indeed, such models are also more accurate on datasets where backgrounds do not match foregrounds. In Figure 5, we observe that a MIXED-RAND-trained model has 17.3% higher accuracy than its ORIGINAL-trained counterpart on MIXED-RAND, and 22.3% higher accuracy on MIXED-NEXT, a dataset where background signals class-consistently mismatch foregrounds. (Recall that MIXED-NEXT images have foregrounds from class $y$ mixed with backgrounds from class $y + 1$, labeled as class $y$.) The MIXED-RAND-trained model also has little variation (at most 3.8%) in accuracy across all five test sets that contain the correct foreground.

Qualitatively, the MIXED-RAND-trained model also appears to place more relative importance on foreground pixels than the ORIGINAL-trained model; the saliency maps of the two models in Figure 6 show that the MIXED-RAND-trained model's saliency maps highlight more foreground pixels than those of ORIGINAL-trained models.

**A fine grained look at dependence on backgrounds.** We now analyze models' reliance on backgrounds at an image-by-image level and ask: for which images does introducing backgrounds help or hurt classifiers' performance? To this end, for each image in ORIGINAL, we decompose

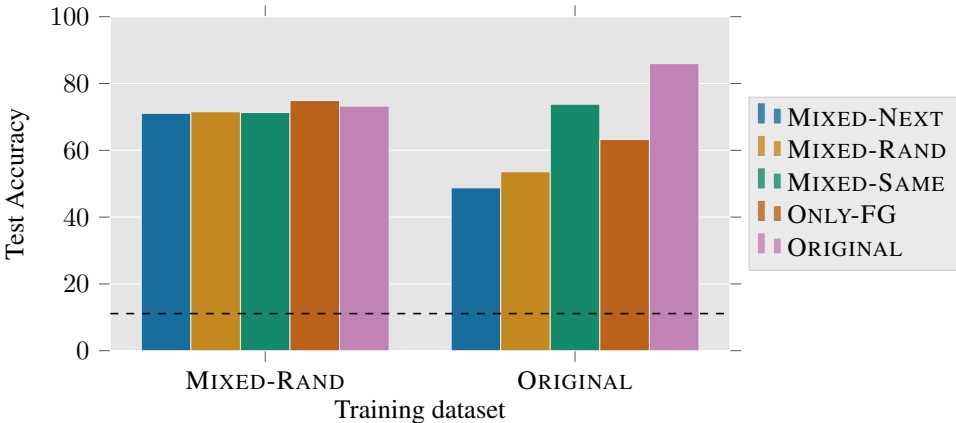

Figure 5: We compare the test performance of a model trained on the synthetic MIXED-RAND dataset with a model trained on ORIGINAL. We evaluate these models on variants of IN-9 that contain identical foregrounds. For the ORIGINAL-trained model, test performance decreases significantly when the background signal is modified during testing. However, the MIXED-RAND-trained model is robust to background changes, albeit at the cost of lower accuracy on images from ORIGINAL.

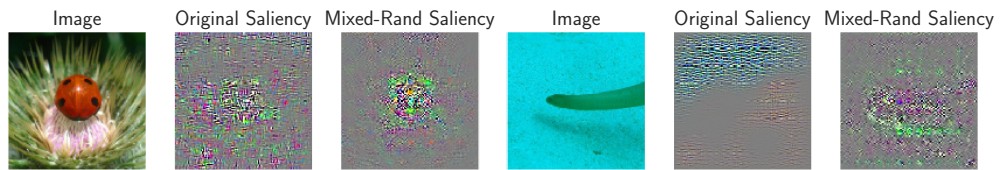

Figure 6: Saliency maps for the the ORIGINAL and MIXED-RAND models on two images. As expected, the MIXED-RAND model appears to place more importance on foreground pixels.

how models use foreground and background signals by examining classifiers' predictions on the corresponding image in MIXED-RAND and ONLY-BG-T. Here, we use the MIXED-RAND and ONLY-BG-T predictions as a proxy for which class the foreground and background signals (alone) point towards, respectively. We categorize each image based on how its background and foreground signals impact classification; we list the categories in Table 3 and show the counts for each category as a histogram per classifier in Figure 7. Our results show that while few backgrounds induce misclassification (see Appendix H for examples), a large fraction of images require backgrounds for correct classification—approximately 35% on the ORIGINAL trained classifiers, as calculated by combining the "BG Required" and "BG+FG Required" categories.

**Further insights derived from IN-9 are discussed in the Appendix D.** We focus on key findings in this section, but also include more comprehensive results and examples of other questions that can be explored by using the toolkit of IN-9 in the Appendix.

## 4  BENCHMARK PROGRESS AND BACKGROUND DEPENDENCE

In the previous sections, we demonstrated that standard image classification models exploit signals from backgrounds. Considering that these models result from progress on standard computer vision benchmarks, a natural question is: *to what extent have improvements on image classification benchmarks resulted from exploiting background correlations?* And relatedly, *how has model robustness to misleading background signals evolved over time?*

As a first step towards answering these questions, we study the progress made by ImageNet models on our synthetic IN-9 dataset variations. In Figure 8 we plot accuracy on our synthetic datasets against ImageNet accuracy for each of the architectures considered. As evidenced by the lines of best fit in Figure 8, accuracy increases on the original ImageNet benchmark generally correspond to

Table 3: Prediction categories we study for a given image-model pair. For a given image, a model can make differing predictions based on the presence or absence of its foreground/background. We label each possible case based on how the background classification relates to the full image classification and the foreground classification. To proxy classifying full images, foregrounds, and backgrounds separately, we classify ORIGINAL, MIXED-RAND, and ONLY-BG-T (respectively). "BG Irrelevant" demarcates images where the foreground classification result is the same as that of the full image (in terms of correctness). We show illustrative examples of BG Required and BG+FG Required below.

| Label | Correct Prediction on Full Image | Correct Prediction on Foreground | Correct Prediction on Background |
|---|---|---|---|
| BG Required | ✓ | ✗ | ✓ |
| BG Fools | ✗ | ✓ | ✗ |
| BG+FG Required | ✓ | ✗ | ✗ |
| BG+FG Fools | ✗ | ✓ | ✓ |
| BG Irrelevant | ✓/✗ | ✓/✗ | — |

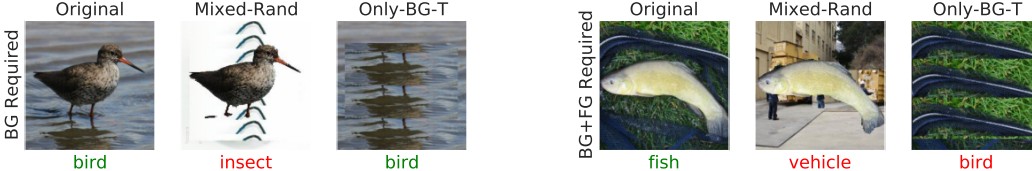

accuracy increases on all of the synthetic datasets. This includes the ONLY-BG datasets—indicating that models *do* improve at extracting correlations from image backgrounds.

Indeed, the ONLY-BG trend observed in Figure 8 suggests that either (a) image classification models can only attain their reported accuracies in the presence of background signals; or (b) these models carry an implicit bias towards features in the background, as a result of optimization technique, model class, etc.—in this case, we may need explicit regularization (e.g., through distributionally robust optimization (Sagawa et al., 2020) or related techniques) to obtain models invariant to these background features. The ONLY-BG trend does not indicate that models are failing per se; it could also indicate that models learn to depend on backgrounds because they are necessary for correctly classifying certain images due to quirks in the ImageNet dataset.

Still, models' *relative* improvement in accuracy across dataset variants is promising—models improve on classifying ONLY-BG-T at a slower (absolute) rate than MIXED-RAND, MIXED-SAME and MIXED-NEXT. Furthermore, the performance gap between the MIXED datasets and the others (most notably, between MIXED-RAND and MIXED-SAME; between MIXED-NEXT and MIXED-RAND; and consequently between MIXED-NEXT and MIXED-SAME) trends towards closing, indicating that models not only are becoming better at using foreground features, but also are becoming more robust to misleading background features (MIXED-RAND and MIXED-NEXT). Finally, models also improve in accuracy faster on NO-FG (which has foreground shape but no texture) than on ONLY-BG-T, which implies that better models are using foreground shape features more effectively.

Overall, the accuracy trends observed from testing ImageNet models on our synthetic datasets reveal that better models (a) are capable of exploiting background correlations, but (b) are increasingly robust to changes in background, suggesting that invariance to background features may not necessarily come at the cost of benchmark accuracy.

## 5 RELATED WORK

Prior works on contextual bias from image backgrounds[4] show that background correlations can be predictive (Torralba, 2003) and can influence model decisions. Zhang et al. (2007) find that (a) a bag-

---

[4]Here, we highlight works analyzing contextual bias from image backgrounds specifically, and discuss contextual bias generally and foreground segmentation in Appendix E.

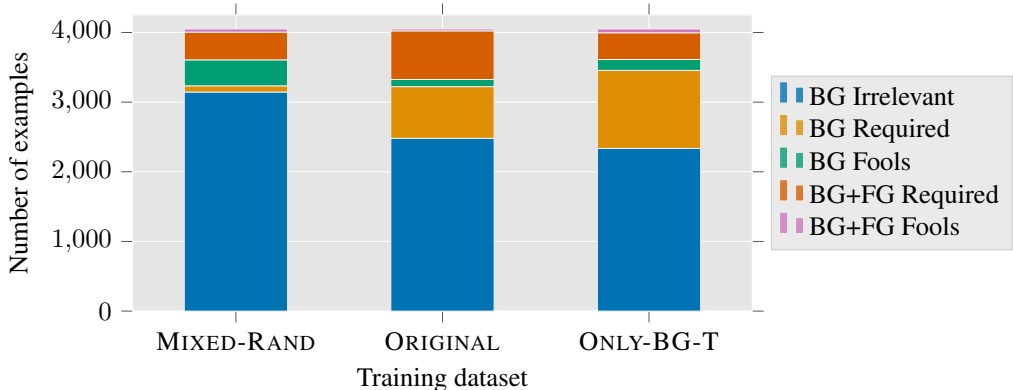

Figure 7: We categorize each test set image based on how a model classifies the full image, the background alone, and the foreground alone (cf. Table 3). The model trained on ORIGINAL needs the background for correct classification on 35% of images (measured by adding "BG Required" and "BG+FG Required"), while a model trained on MIXED-RAND is much less reliant on background. The model trained on ONLY-BG-T requires the background most, as expected; however, the model often misclassifies both the full image and the background, so the "BG Irrelevant" subset is still sizable.

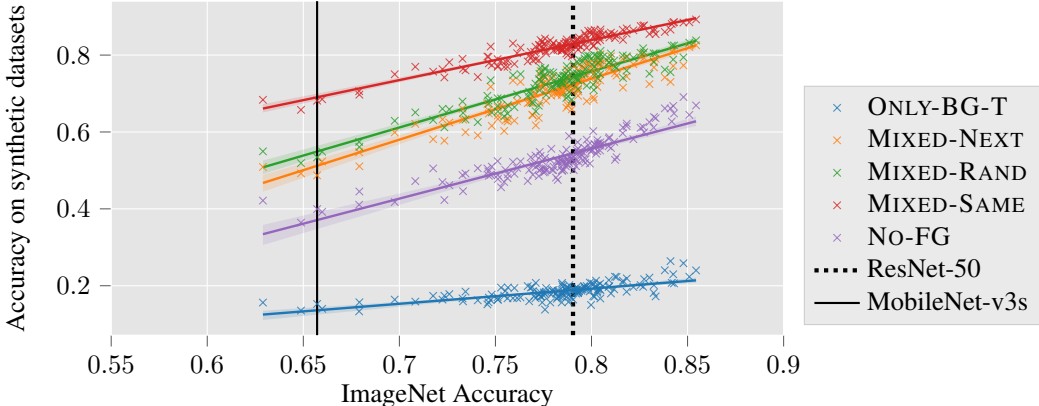

Figure 8: Measuring progress on each of the synthetic ImageNet-9 datasets with respect to progress on the standard ImageNet test set. Higher accuracy on ImageNet generally corresponds to higher accuracy on each of the constructed datasets, but the rate at which accuracy grows varies based on the types of features present in each dataset. Each pre-trained model corresponds to a vertical line on the plot—we mark ResNet-50 and MobileNet-v3s models for reference.

of-features object detection algorithm depends on image backgrounds in the PASCAL dataset and (b) using this algorithm on a training set with varying backgrounds leads to better generalization. Beery et al. (2018) and Barbu et al. (2019) collect new test datasets of animals and objects, respectively. Barbu et al. (2019) focus on object classes that also exist in ImageNet, and their new test set contains objects photographed in front of unconventional backgrounds and in unfamiliar orientations. Both works show that computer vision models experience significant accuracy drops when trained on data with one set of backgrounds and tested on data with another. Sagawa et al. (2020) create a synthetic dataset of Waterbirds, where waterbirds and landbirds from one dataset are combined with water and land backgrounds from another. They show that a model's reliance on spurious correlations with the background can be harmful for small subgroups of data where those spurious correlations no longer hold (e.g. landbirds on water backgrounds). Rosenfeld et al. (2018) analyze background dependence for object detection (as opposed to classification) models on the MS-COCO dataset. They transplant an object from one image to another image, and find that object detection models may detect the transplanted object differently depending on its location, and that the transplanted object may also cause mispredictions on other objects in the image. Zech et al. (2018) study medical imaging and

show that a model learned to detect a hospital-specific metal token on medical scans. The model then used each hospital's pneumonia prevalence rate to predict pneumonia fairly well (without learning much about actually detecting pneumonia).

The only work that, similarly to us, studies a large-scale dataset in this context is Zhu et al. (2017), who analyze ImageNet and show that AlexNet can achieve nontrivial accuracy on a dataset similar to our ONLY-BG-B dataset. While sufficient for establishing that backgrounds can be used for classification, this dataset also introduces biases by adding large black rectangular patches to all of the images (which our ONLY-BG-T dataset fixes). In comparison to Zhu et al. (2017) and the other prior works, we: (a) properly segment foregrounds and backgrounds using the GrabCut algorithm instead of relying on rectangular bounding boxes; (b) create dataset variations that allow us to measure not just model performance without foregrounds, but also the relative influence of foregrounds and backgrounds on model predictions; (c) control for the effect of image artifacts by focusing on comparisons between the MIXED-SAME and MIXED-RAND datasets; (d) study model robustness to adversarial backgrounds; (e) study a larger and more recent set of classifiers (He et al., 2016; Zagoruyko & Komodakis, 2016; Tan & Le, 2019); (f) show how improvements they give on ImageNet relate to background dependence; and (g) make our benchmarking toolkit publicly accessible for others to use and build on.

## 6    DISCUSSION AND CONCLUSION

In this work, we study the extent to which classifiers rely on image backgrounds. To this end, we create a toolkit for measuring the precise role of background and foreground signal that involves constructing new test datasets that contain different amounts of each. Through these datasets we establish both the usefulness of background signal and the tendency of our models to depend on backgrounds, even when relevant foreground features are present. Our results show that our models are not robust to changes in the background, either in the adversarial case, or in the average case.

As most ImageNet images have human-recognizable foreground objects, our models appear to rely on background more than humans on that dataset. The fact that models can be fooled by adversarial background changes on 88% of all images highlights how poorly computer vision models may perform in an out-of-distribution setting. However, contextual information like the background can still be useful in certain settings. After all, humans do use backgrounds as context in visual processing, and the background may be necessary if the foreground is blurry or distorted (Torralba, 2003). Therefore, reliance on background is a nuanced question that merits further study.

On one hand, our findings provide evidence that models succeed by using background correlations, which may be undesirable in some applications. On the other hand, we find that advances in classifiers have given rise to models that use foregrounds more effectively and are more robust to changes in the background. To obtain even more robust models, we may want to draw inspiration from successes in training on the MIXED-RAND dataset (a dataset designed to neutralize background signal—cf. Table 1), related data-augmentation techniques (Shetty et al., 2019), and training algorithms like distributionally robust optimization (Sagawa et al., 2020) and model-based robust learning (Robey et al., 2020). Overall, the toolkit and findings in this work help us to better understand models and to monitor our progress toward the goal of reliable machine learning.

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

## A  DATASETS DETAILS

We choose the following 9 high-level classes.

| Class | WordNet ID | Number of sub-classes |
|---|---|---|
| **Dog** | n02084071 | 116 |
| **Bird** | n01503061 | 52 |
| **Vehicle** | n04576211 | 42 |
| **Reptile** | n01661091 | 36 |
| **Carnivore** | n02075296 | 35 |
| **Insect** | n02159955 | 27 |
| **Instrument** | n03800933 | 26 |
| **Primate** | n02469914 | 20 |
| **Fish** | n02512053 | 16 |

Table 4: The 9 classes of ImageNet-9.

All datasets used in the paper are balanced by randomly removing images from classes that are over-represented. We only keep as many images as the smallest post-modification synthetic dataset, so all synthetic datasets (except IN-9L) have the same number of images. We also use a custom GUI to manually process the test set to improve data quality. For IN-9L, the only difference from using the corresponding classes in the original ImageNet dataset is that we balance the dataset.

**For all images**: we apply the following filters before adding each image to our datasets.

- The image must have bounding box annotations.
- For simplicity, each image must have exactly one bounding box. A large majority of images that have bounding box annotations satisfy this.

**For images needing a properly segmented foreground**: This includes the 3 MIXED datasets, ONLY-FG, and NO-FG. We filter out images based on the following criteria.

- Because images are cropped before they are fed into models, we require that less than 50% of the bounding box is removed by the crop, to ensure that the foreground still exists. Almost all images pass this filter.
- The OpenCV foreground segmentation function `cv2.grabCut` (used to extract the foreground shape) must work on the image. We remove images where it fails.
- For the test set only, we manually remove images with foreground segmentations that retain a significant portion of the background signal.
- For the test set only, we manually remove foreground segmentations that are very bad (e.g. the segmentation selects part of the image, and that part doesn't contain the foreground object).

**For images needing only background signal**: This includes ONLY-BG-B and ONLY-BG-T. In this case, we apply the following criteria:

- The bounding box must not be too big (more than 90% of the image). The intent here is to avoid ONLY-BG-B images being just a large black rectangle.
- For the test set only, we manually remove ONLY-BG images that still have an instance of the class even after removing the bounding box. This occurs when the bounding boxes are imperfect or incomplete (e.g. only one of two dogs in an image is labeled with a bounding box).

**Creating the ONLY-BG-T dataset**: We first make a "tiled" version of the background by finding the largest rectangular strip (horizontal or vertical) outside the bounding box, and tiling the entire

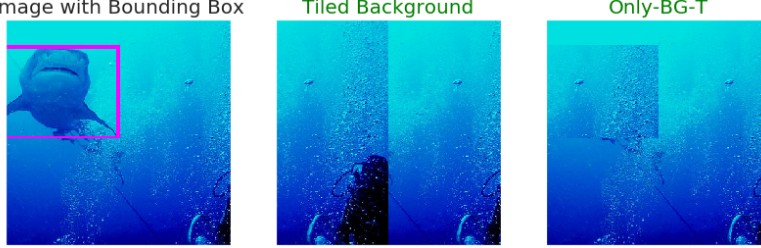

Figure 9: Visualization of how ONLY-BG-T is created.

image with that strip. We then replace the removed foreground with the tiled background. A visual example is provided in Figure 9. We purposefully choose not to use deep-learning-based inpainting techniques such as (Shetty et al., 2018) to replace the removed foreground, as such methods could lead to biases that the inpainting model has learned from the data. For example, an inpainting model may learn that the best way to inpaint a missing chunk of a flower is to place an insect there, which is something we want to avoid.

**Motivation for each IN-9 variation:** We create ONLY-BG-B and ONLY-BG-T to remove the foreground completely, including the shape of the foreground object. We intend for ONLY-BG-B to be directly comparable to the prior work of Zhu et al. (2017) that uses similar methodology to evaluate older AlexNet models, while ONLY-BG-T is a more natural-looking background that avoids black rectangles introduced in ONLY-BG-B.

The NO-FG dataset is created to retain the foreground shape, but not the texture. We can use it to assess the relative importance of foreground shape compared to foreground texture.

Finally, we create four datasets that have identical foregrounds but each have distinct background signals. ONLY-FG has a pure black background to go with the foreground. MIXED-SAME has background signal from the same class as the foreground. MIXED-RAND has background signal from a random class, so it can be thought of as having neutral background signal. MIXED-NEXT has background signal from the next class, which will always be in conflict with the foreground . Any artifacts in the foreground that result from our image processing pipeline are equally present in all four datasets. Thus, these datasets help to isolate how much backgrounds alone influence model predictions *when the correct foreground exists in the image*.

**Full-ImageNet version of each synthetic variation:** We also apply the same methodology for disentangling foreground and background signal to the entire ImageNet validation set, creating Full-ImageNet (Full-IN) versions of each of our 7 dataset variations.

We evaluate a pre-trained ResNet-50 on Full-IN for comparison in Table 5, and observe similar trends to ImageNet-9 that lead to similar conclusions on model background reliance. We choose to focus on ImageNet-9 results in the main paper because of the following shortcomings of Full-IN.

1. Individual classes are quite small, as some classes have very few (or even zero) images that make it through our filters due to lack of proper annotated bounding boxes.

2. When bounding boxes do exist, their quality is often lower than those in the IN-9 classes. For example, many images of fruit contain multiple fruit, but only one will be properly annotated with a bounding box.

3. When creating the MIXED-NEXT equivalent for Full-IN, the next class is often similar to the previous one. For example, many dog breeds occur consecutively in ImageNet. Thus, Full-IN's MIXED-NEXT frequently has backgrounds that are similar to backgrounds from the foreground class.

# B    Explaining the Decreased BG-Gap of pre-trained ImageNet models

We investigate two possible explanations for why pre-trained ImageNet models have a smaller BG-Gap than models trained on ImageNet-9. Understanding this phenomenon can help inform how models should be trained to be more background-robust. We find slight improvements to background-robustness from training on more fine-grained classes. We find that training on larger datasets helps only slightly when the training dataset set size is smaller than IN-9L, but larger improvements occur when the training dataset size is bigger. Thus, we encourage training on larger datasets if reduced background robustness is the goal.

## B.1    The Effect of Fine-grainedness on the BG-Gap

One possible explanation is that training models to distinguish between finer-grained classes forces them to focus more on the foreground, which contains relevant features for making those fine-grained distinctions, than the background, which may be fairly similar across sub-classes of a high-level class. This suggests that asking models to solve more fine-grained tasks could improve model robustness to background changes.

To test the effect of fine-grainedness on ImageNet-9, we make a related dataset called IN-9LB that uses the same 9 high-level classes and can be cleanly modified into more fine-grained versions. Specifically, for IN-9LB we choose exactly 16 sub-classes for each high-level class, for a total of 144 ImageNet classes. To create successively more fine-grained versions of the IN-9LB dataset, we group every $n$ sub-classes together into a higher-level class, for $n \in \{1, 2, 4, 8, 16\}$. Here, $n = 1$ corresponds to keeping all 144 ImageNet classes as they are, while $n = 16$ corresponds to only having 9 high-level classes, like ImageNet-9. Because we keep all images from those original ImageNet classes, this dataset is the same size as IN-9L.

We train models on IN-9LB at different levels of fine-grainedness and evaluate the BG-Gap of those models in Figure 10. We find that fine-grained models have a smaller BG-Gap as well as better performance on Mixed-Next, but the improvement is very slight and also comes at the cost of decreased accuracy on Original. The BG-Gap of the most fine-grained classifier is 2.3% smaller than the BG-Gap of the most coarse-grained classifier, showing that fine-grainedness does improve background-robustness. However, the improvement is still small compared to the size of the BG-Gap (which is 13.3% for the fine-grained classifier).

## B.2    The Effect of Larger Dataset Size on the BG-Gap

A second possible explanation for why pre-trained ImageNet models have a smaller BG-Gap is that training on larger datasets is important for background-robustness. To evaluate this possibility, we train models on different-sized subsets of IN-9LB. The largest dataset we train on is the full IN-9LB dataset, which is 4 times as large as IN-9, and the smallest is 1/4 as large as IN-9. Figure 11 shows that increasing the dataset size does increase overall performance but only slightly decreases the BG-Gap.

Next, we train models on different-sized subsets of ImageNet; we use the pre-trained ResNet-50 ImageNet model for full-sized ImageNet, and we train new ResNet-50 models on subsets that are 1/2, 1/4, 1/8, 1/16, and 1/32 as large as ImageNet. In these cases, we observe in Figure 12 that training on more data does not help significantly when the training dataset sizes are still small, but it does help more noticeably for models trained on 1/2 of ImageNet and all of ImageNet.

It is possible that having both a fine-grained class structure and more training data simultaneously is important for background-robustness. Furthermore, more training data (from other classes that are not in IN-9L) may also be the cause of the increased background-robustness of pre-trained ImageNet models.

## B.3    Summary of methods investigated to reduce the BG-Gap

In Figure 13, we compare the BG-Gap of ResNet-50 models trained on different datasets and with different methods to a ResNet-50 pre-trained on ImageNet. We explore $\ell_p$-robust training, increasing

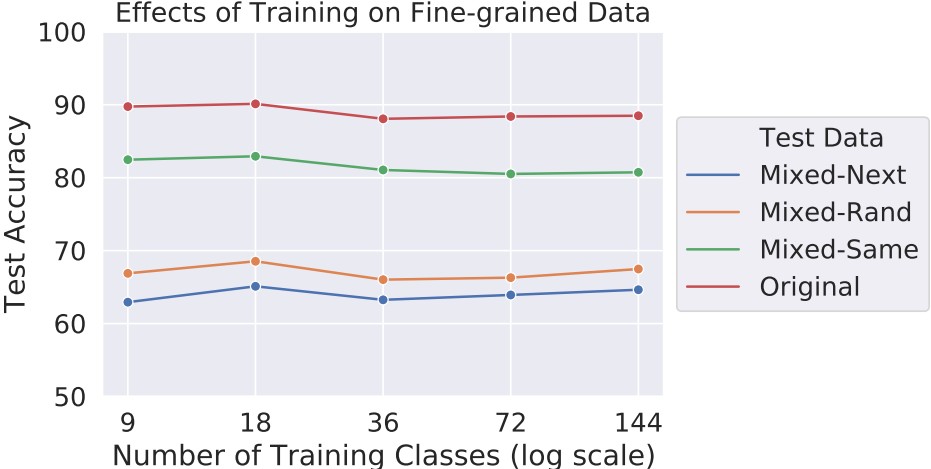

Figure 10: We train models on IN-9LB at different levels of fine-grainedness (more training classes is more fine-grained). The BG-GAP, or the difference between the test accuracies on MIXED-SAME and MIXED-RAND, decreases as we make the classification task more fine-grained, but the decrease is small compared to the size of the BG-GAP.

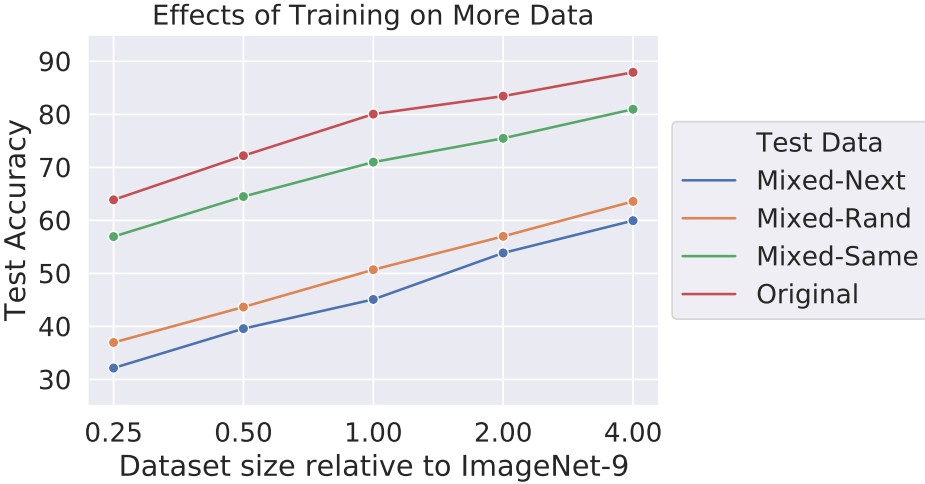

Figure 11: We train models on different-sized subsets of IN-9LB. The largest training set we use is the full IN-9LB dataset, which is 4 times larger than ImageNet-9. While performance on all test datasets improves as the amount of training data increases, the BG-GAP has almost the same size regardless of the amount of training data used.

dataset size, and making the classification task more fine-grained, and find that none of these methods reduces the BG-GAP as much as pre-training on ImageNet. The only method that reduces the BG-GAP significantly more is training on MIXED-RAND. Furthermore, the same trends hold true if we measure the difference between MIXED-SAME and MIXED-NEXT as opposed to the BG-GAP (the difference between MIXED-SAME and MIXED-RAND).

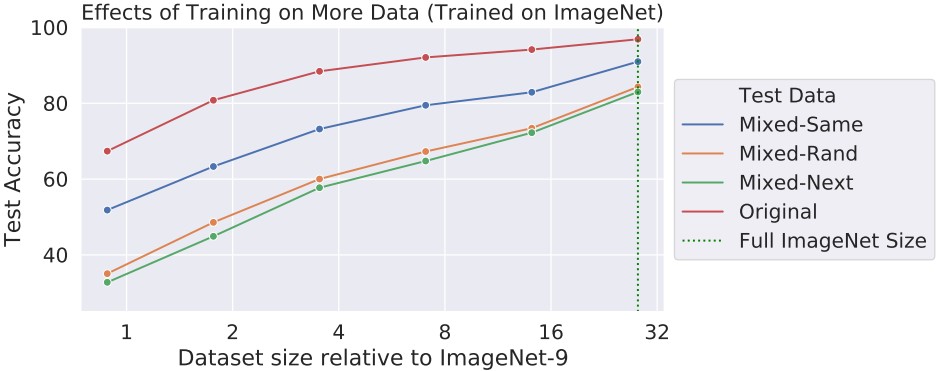

Figure 12: We train models on different-sized subsets of ImageNet. We use a pre-trained ResNet-50 for the rightmost datapoints corresponding to training on the full ImageNet dataset, which is about 30 times larger than ImageNet-9. The BG-GAP begins to decrease when the training dataset set size is sufficiently large.

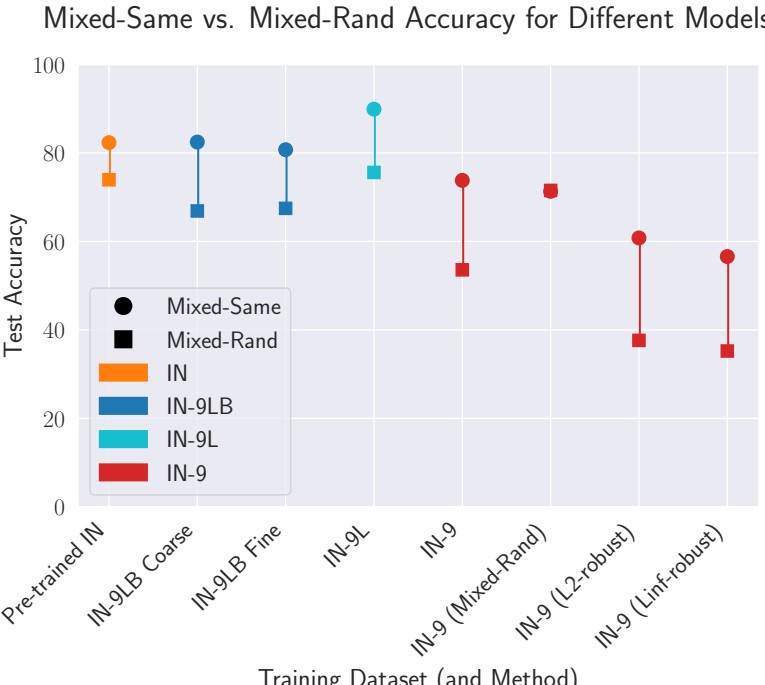

Figure 13: We compare various different methods of training models and measure their BG-GAP, or the difference between MIXED-SAME and MIXED-RAND test accuracy. We find that (1) Pre-trained IN models have surprisingly small BG-GAP. (2) Increasing fine-grainedness (IN-9LB Coarse vs. IN-9LB Fine) and dataset size (IN-9 vs. IN-9L) decreases the BG-GAP only slightly. (3) $\ell_p$-robust training does not help. (4) Training on MIXED-RAND (cf. Section 3 appears to be the most effective strategy for reducing the BG-GAP. For such a model, the MIXED-SAME and MIXED-RAND accuracies are nearly identical.

## C   TRAINING AND EVALUATION DETAILS

For all models, we use fairly standard training settings for ImageNet-style models. We train for 200 epochs using SGD with a batch size of 256, a learning rate of 0.1 (with learning rate drops every 50 epochs), a momentum parameter of 0.9, a weight decay of $1e-4$, and data augmentation (random resized crop, random horizontal flip, and color jitter). Unless specified, we always use a standard ResNet-50 architecture (He et al., 2016). For the experiment depicted in Figure 11, we found that using a smaller learning rate of 0.01 was necessary for training to converge on the smallest training sets. Thus, we used that same learning rate for all models in Figure 11.

When evaluating ImageNet classifiers on IN-9, we map all ImageNet predictions to their corresponding coarse-grained class in IN-9. For example, we map both `giant schnauzer` and `Irish terrier` to dog, and both `goldfish` and `tiger shark` to FISH. If an ImageNet classifier outputs a class that has no corresponding coarse-grained class in IN-9, we consider the prediction incorrect.

## D   ADDITIONAL EVALUATION RESULTS

We include full results of training models on every synthetic IN-9 variation and then testing them on every synthetic IN-9 variation in Table 5. In addition to being more comprehensive, this table and these IN-9 variations can help answer a variety of questions, of which we provide three examples here. Finally, we also evaluate a pre-trained model on Full-ImageNet (Full-IN) versions of each synthetic IN-9 variation for comparison.

**How does more training data affect model performance with and without object shape?**

We already show closely related results on the effect of more training data on the BG-GAP in Figure 11. Here, we compare model test performance on the NO-FG and ONLY-BG-B test sets. Both replace the foreground with black, but only NO-FG retains the foreground shape.

By comparing the models trained on ORIGINAL and IN-9L (4x more training data), we find that

1. The ORIGINAL-trained model performs about 9% better on NO-FG than ONLY-BG-B, indicating that it can slightly improve accuracy by using object shape.

2. The IN-9L-trained model performs about 22% better on NO-FG than ONLY-BG-B, showing that it can improve accuracy far more by using object shape.

Furthermore, both models perform very similarly on ONLY-BG-B. Thus, this suggests that more training data may allow models to learn to use object shape more effectively. Understanding this phenomena further could help inform model training and dataset collection if the goal is to train models that are able to leverage shape effectively.

**How much information is leaked from the size of the foreground bounding box?**

The scale of an object already gives signal correlated with the object class (Torralba, 2003). Even though they are designed to avoid having foreground signal, the background-only datasets ONLY-BG-B and ONLY-BG-T may inadvertently leak information about object scale due to the bounding box sizes being recognizable.

To gauge the extent of this leakage, we can measure how models trained on datasets where only the foreground signal has useful correlation (MIXED-RAND or ONLY-FG) perform on the background-only test sets. We find that there is small signal leakage from bounding box size alone—a model trained on ONLY-FG achieves about 23% background-only test accuracy, suggesting that it is able to exploit the signal leakage to some degree. A model trained on MIXED-RAND achieves about 15% background-only test accuracy, just slightly better than random, perhaps because it is harder for models to measure (and thus, make use of) object scale when training on MIXED-RAND.

The existence of a small amount of information leakage in this case shows the importance of comparing MIXED-SAME (as opposed to just ORIGINAL) with MIXED-RAND and MIXED-NEXT when assessing model dependence on backgrounds. Indeed, the MIXED datasets may contain (1) image processing artifacts, such as rough edges from the foreground processing, and (2) small traces

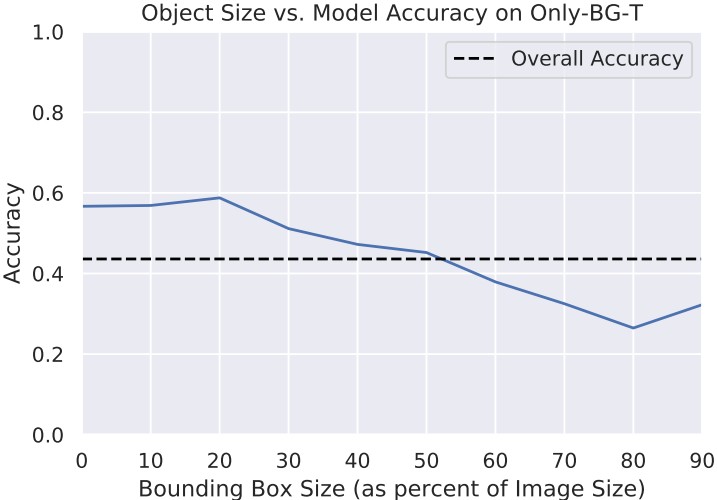

Figure 14: Comparing model accuracy on ONLY-BG-T across different foreground object bounding box sizes. We observe that the model is more likely to succeed when shown only image backgrounds if the removed foreground objects have smaller bounding boxes. The dotted line represents the overall accuracy of the model on ONLY-BG-T (averaged over all bounding box sizes).

of the original background. This makes it important to control for both factors when measuring how models react to varying background signal.

**How does foreground bounding box size affect accuracy on ONLY-BG-T?**

We further find that models are more able to predict accurately using the background signal alone when the foreground object is smaller—this is visualized in Figure 14. Intuitively this result makes sense, as most state-of-the-art models are trained with cropping-based data augmentation, which can remove small foreground objects from training images. Thus, models are actually trained to succeed when small foreground objects are cropped out, and our toolkit confirms that this is indeed the case.

| Trained on | Test Dataset | | | | | | | | |
|---|---|---|---|---|---|---|---|---|---|
| | MIXED-NEXT | MIXED-RAND | MIXED-SAME | NO-FG | ONLY-BG-B | ONLY-BG-T | ONLY-FG | ORIGINAL | IN-9L |
| MIXED-NEXT | 78.07 | 53.28 | 48.49 | 16.20 | 11.19 | 8.22 | 59.60 | 52.32 | 46.44 |
| MIXED-RAND | 71.09 | 71.53 | 71.33 | 26.72 | 15.33 | 14.62 | 74.89 | 73.23 | 67.53 |
| MIXED-SAME | 45.41 | 51.36 | 74.40 | 39.85 | 35.19 | 41.58 | 61.65 | 75.01 | 69.21 |
| NO-FG | 13.70 | 18.74 | 42.79 | 70.91 | 36.79 | 42.52 | 31.48 | 48.94 | 47.62 |
| ONLY-BG-B | 10.35 | 15.41 | 38.37 | 37.85 | 54.30 | 42.54 | 21.38 | 42.10 | 41.01 |
| ONLY-BG-T | 11.48 | 17.09 | 45.80 | 40.84 | 38.49 | 50.25 | 19.19 | 49.06 | 47.94 |
| ONLY-FG | 33.04 | 35.88 | 47.63 | 27.90 | 23.58 | 22.59 | 84.20 | 54.62 | 51.50 |
| ORIGINAL | 48.77 | 53.58 | 73.80 | 42.22 | 32.94 | 40.54 | 63.23 | 85.95 | 80.38 |
| IN-9L | 71.21 | 75.60 | 89.90 | 55.78 | 34.02 | 43.60 | 84.12 | 96.32 | 94.61 |
| ImageNet | 82.99 | 84.32 | 90.99 | 52.69 | 12.69 | 17.36 | 90.17 | 96.89 | 95.33 |
| ImageNet (Full-IN) | 51.47 | 48.69 | 64.34 | 21.70 | 7.98 | 9.51 | 59.19 | 76.07 | - |

Table 5: The test accuracies, in percentages, of ResNet-50 models trained on all variants of ImageNet-9, and a pre-trained ImageNet ResNet-50. The bottom row and the second-to-last-row test the same pre-trained ImageNet model; however, the bottom row tests the model on the Full-IN version of each dataset variation. Testing on Full-IN shows similar trends as testing on ImageNet-9. Note that the MIXED-NEXT test accuracy is actually higher than the MIXED-RAND test accuracy in the bottom row because the next class is often very similar to the previous class in Full-IN.

**What about other ways of modifying the background signal?**

One can modify the background in various other ways—for example, instead of replacing the background with black as in ONLY-FG, the background can be blurred as in the BG-BLURRED image of Figure 15. As expected, blurred backgrounds are still slightly correlated with the correct class. Thus, test accuracies for standard models on this dataset are higher than on ONLY-FG, but lower than on MIXED-SAME (which has signal from random class-aligned backgrounds that are *not* blurred). While we do not investigate all possible methods of modifying background signal,

Original        Only-FG        BG-Blurred

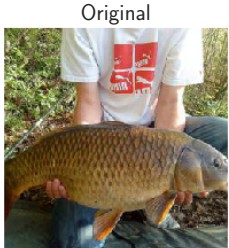 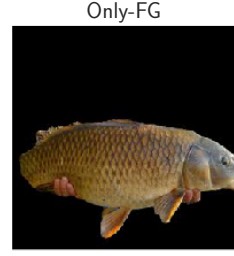 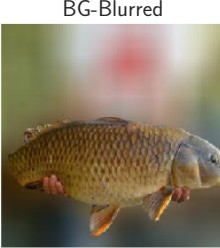

Figure 15: Backgrounds can also be modified in other ways; for example, it can be blurred. Our evaluations on this dataset show similar results.

we believe that the variations we do examine in ImageNet-9 already improve our understanding of how background signals matter. Investigating other variations could provide an even more nuanced understanding of what parts of the background are most important.

## E ADDITIONAL RELATED WORKS AND EXPLICIT COMPARISONS

There has been prior work on mitigating contextual bias in image classification, the influence of background signals on various datasets, and techniques like foreground segmentation that we leverage.

**Mitigating Contextual Bias:** (Khosla et al., 2012) focuses on mitigating dataset-specific contextual bias and proposes learning SVMs with both general weights and dataset-specific weights, while (Myung Jin et al., 2012) creates an out-of-context detection task with 209 out-of-context images and suggests using graphical models to solve it. (Shetty et al., 2019) focuses on the role of co-occurring objects as context in the MS-COCO dataset, and uses object removal to show that (a) models can still predict a removed object when only co-occurring objects are shown, and (b) special data-augmentation can mitigate this.

**Explicit Comparison to Prior Works Studying the Influence of Backgrounds**: In comparison to prior works on the influence of image backgrounds (described in Section 5), our work contributes the following.

- We develop a toolkit for analyzing the background dependence of ImageNet classifiers, the most common benchmark for computer vision progress. Only (Zhu et al., 2017), which we compare to in Section 5, also focuses on ImageNet.
- The test datasets we create separate and mix foreground and background signals in various ways (cf. Table 1), allowing us to study the sensitivity of models to these signals in a more fine-grained manner.
- Our toolkit for separating foreground and background can be applied without human-annotated foreground segmentation, which prior works on MS-COCO and Waterbirds rely on. This is important because foreground segmentation annotations are hard to collect and do not exist for ImageNet.
- We study the extent of background dependence in the extreme case of adversarial back-grounds.
- We focus on better vision models, including ResNet (He et al., 2016), Wide ResNet(Zagoruyko & Komodakis, 2016), and EfficientNet (Tan & Le, 2019).
- We evaluate how improvements on the ImageNet benchmark have affected background dependence (cf. Section 4).
- We will publicly release our toolkit (code and datasets) for benchmarking background dependence so that others can also use it to better understand their own models. Our toolkit is compatible with any ImageNet-trained model.

**Foreground Segmentation and Image Inpainting:** In order to create IN-9 and its variants, we rely on OpenCV's implementation of the foreground segmentation algorithm GrabCut (Rother et al.,

2004). Foreground segmentation is a branch of computer vision that seeks to automatically extract the foreground from an image (Harville et al., 2001). After finding the foreground, we remove it and simply replace the foreground with copies of parts of the background. Other works solve this problem, called image inpainting, either using exemplar-based methods (Criminisi et al., 2004) or using deep learning Yu et al. (2018); Shetty et al. (2018). (Shetty et al., 2018) both detects the foreground for removal and inpaints the removed region. However, more advanced inpaintings techniques can be slow and inaccurate when the region that must be inpainted is relatively large (Shetty et al., 2018), which is the case for many ImageNet images. Exploring better ways of segementing the foreground and inpainting the removed foreground could improve our analysis toolkit further.

# F ADDITIONAL EXAMPLES OF SYNTHETIC DATASETS

We randomly sample an image from each class, and display all synthetic variations of that image, as well as the predictions of a pre-trained ResNet-50 (trained on IN-9L) on each variant.

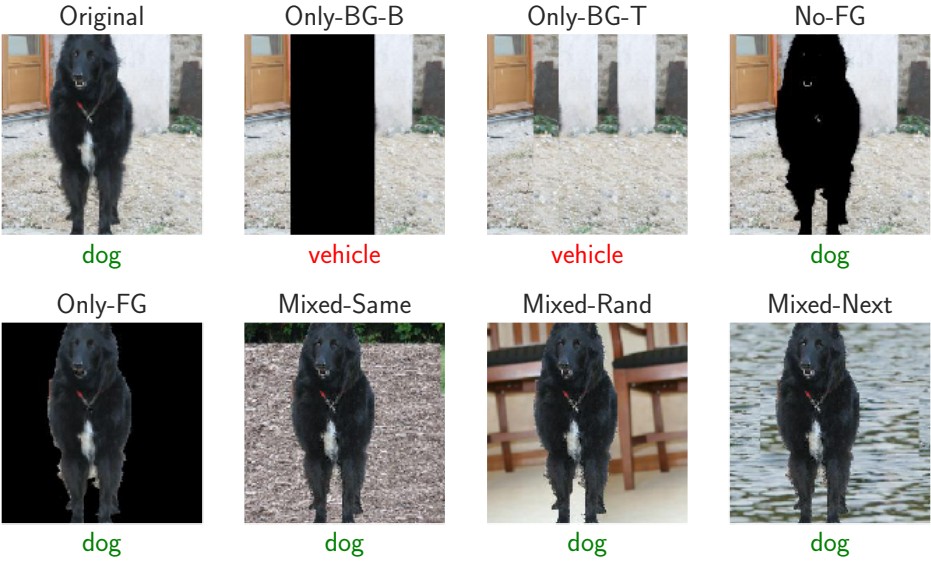

Figure 16: ImageNet-9 variations—Dog.

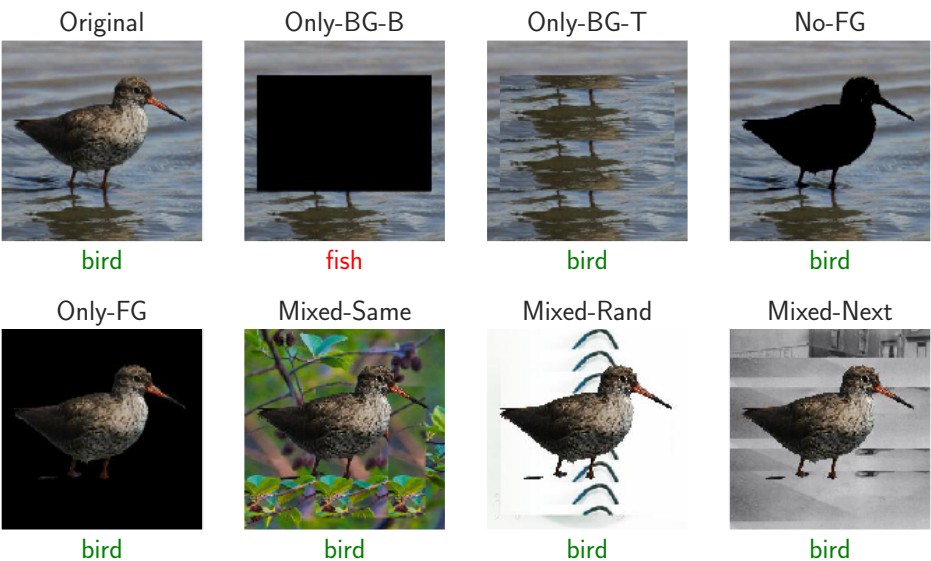

Figure 17: ImageNet-9 variations—Bird.

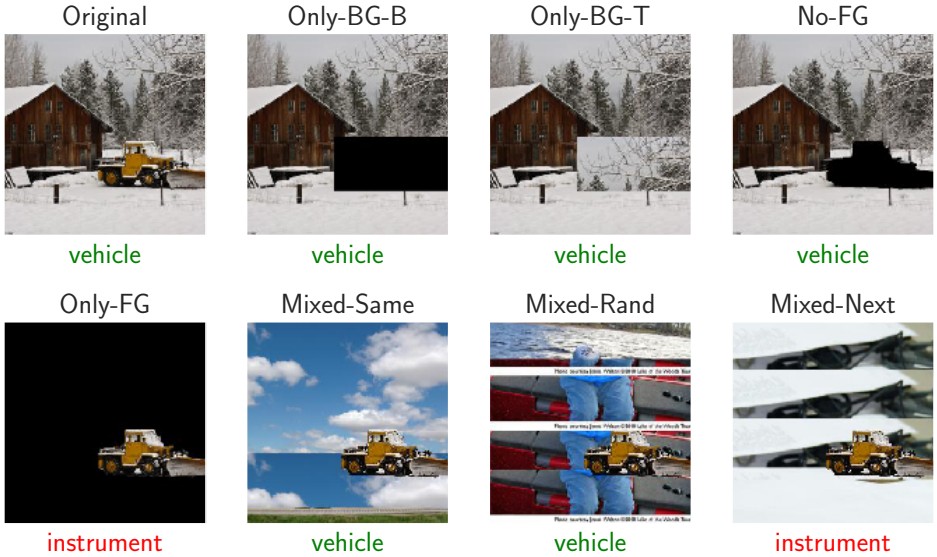

Figure 18: ImageNet-9 variations—Vehicle.

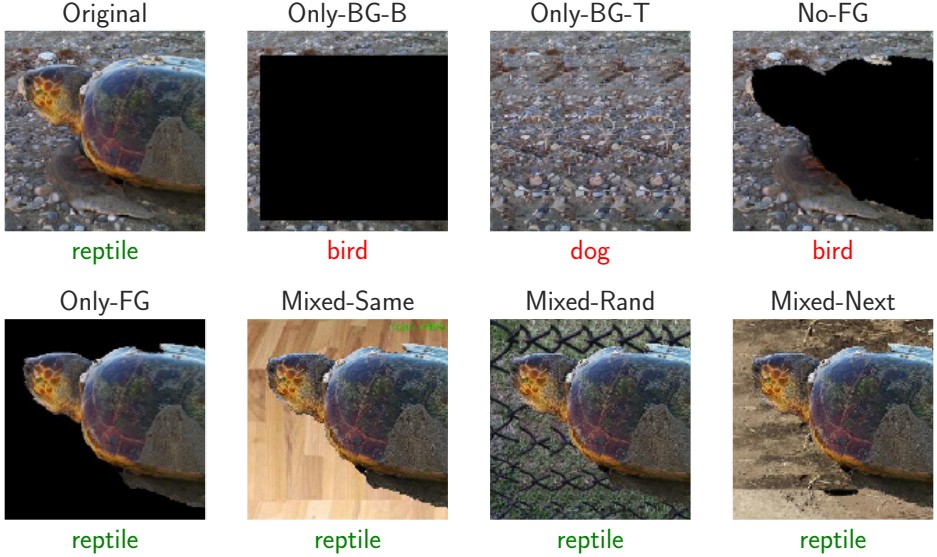

Figure 19: ImageNet-9 variations—Reptile.

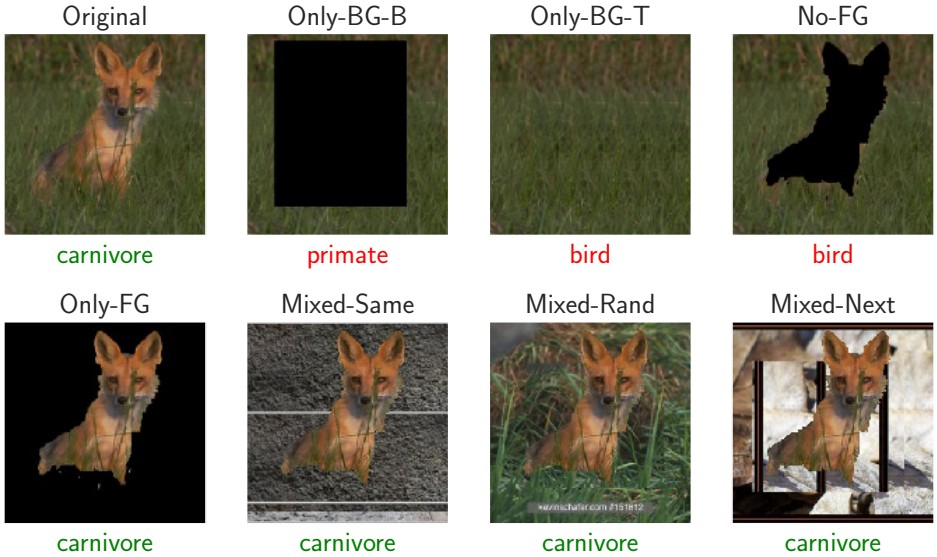

Figure 20: ImageNet-9 variations—Carnivore.

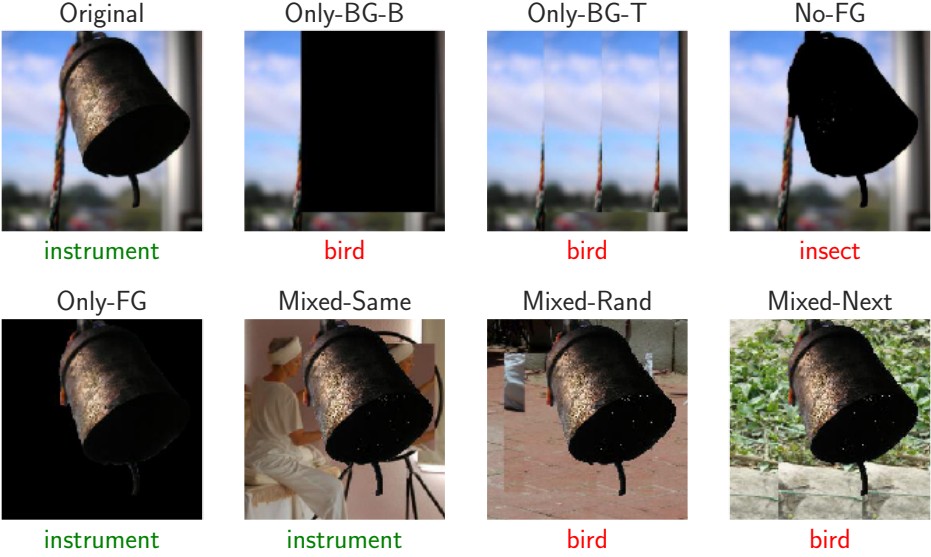

Figure 21: ImageNet-9 variations—Instrument.

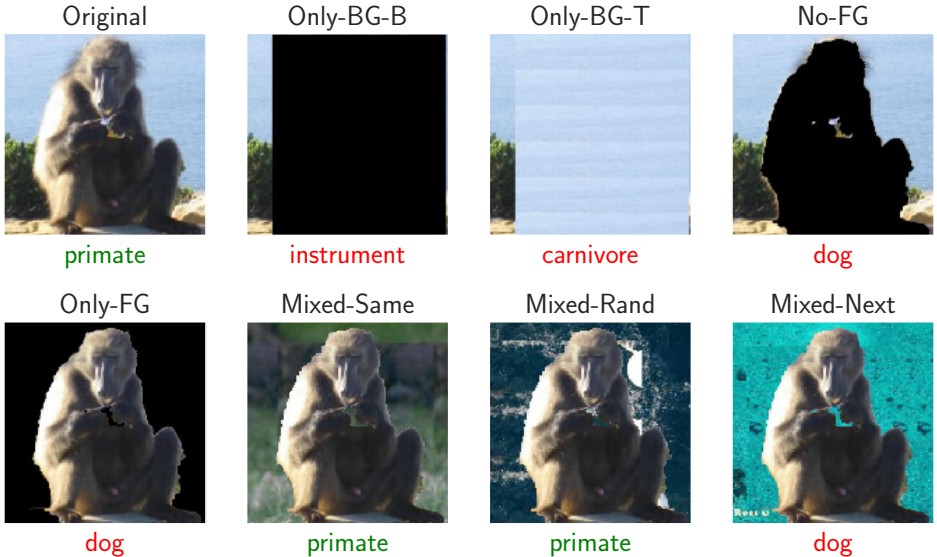

Figure 22: ImageNet-9 variations—Primate.

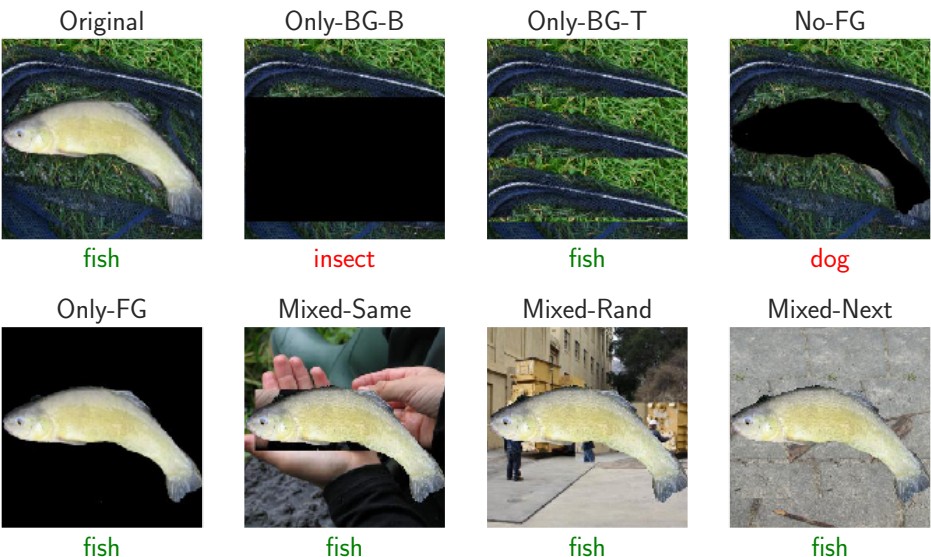

Figure 23: ImageNet-9 variations—Fish.

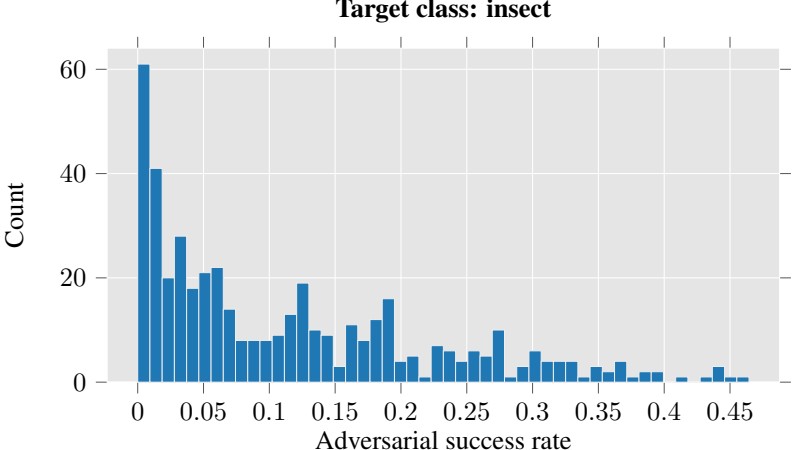

Figure 24: Histogram of insect backgrounds grouped by how often they cause (non-insect) fore-grounds to be classified as insect by a IN-9L-trained model. We visualize the five backgrounds that fool the classifier on the largest percentage of images in Figure 4.

## G  ADVERSARIAL BACKGROUNDS

We compute the adversarial background attack success rate for 4 models in Table 6. While the MIXED-RAND model is more adversarially background robust than the ORIGINAL model, it is less adversarially background robust than the IN-9L model. The model trained on all of ImageNet is the most adversarially background robust of all models. This suggests that increasing training dataset size (IN-9L) has a bigger effect on adversarial background robustness than randomizing backgrounds during training (MIXED-RAND). On the other hand, the MIXED-RAND model has a much lower BG-GAP than the IN-9L model, indicating that models with a smaller BG-GAP are not necessarily robust to adversarial backgrounds, and vice versa.

| Training Dataset | ORIGINAL | MIXED-RAND | IN-9L | ImageNet |
|---|---|---|---|---|
| **Attack Success Rate** | 99.0% | 93.5% | 88.0% | 77.7% |

Table 6: Adversarial backgrounds attack success rates for 4 models analyzed in this paper. The ORIGINAL and the MIXED-RAND are trained on equally small datasets, IN-9L is trained on 4x more data, and the ImageNet model is trained on the most data.

Next, we visualize the attack success rate distribution of the different backgrounds from the insect class in Figure 24. The long tail of the distribution indicates that many backgrounds are especially capable of fooling models.

Finally, we include the 5 most fooling backgrounds for all classes, the fool rate for each of those 5 backgrounds, and the total fool rate across all backgrounds from that class (on the left of each row) below.

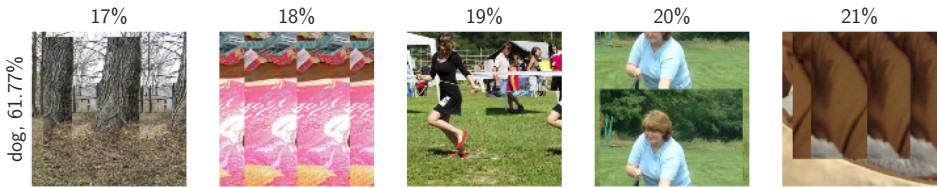

Figure 25: Most adversarial backgrounds—Dog.

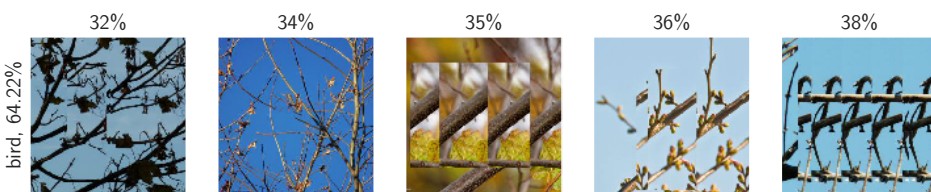

Figure 26: Most adversarial backgrounds—Bird.

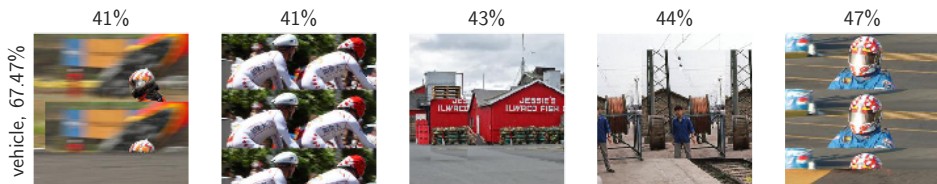

Figure 27: Most adversarial backgrounds—Vehicle.

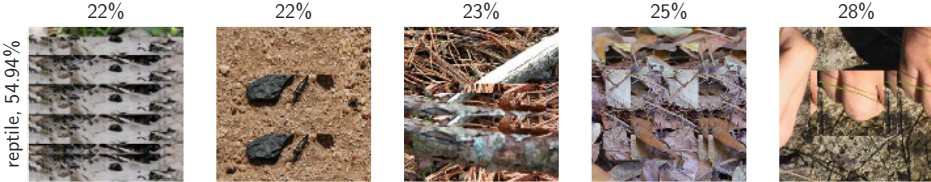

Figure 28: Most adversarial backgrounds—Reptile.

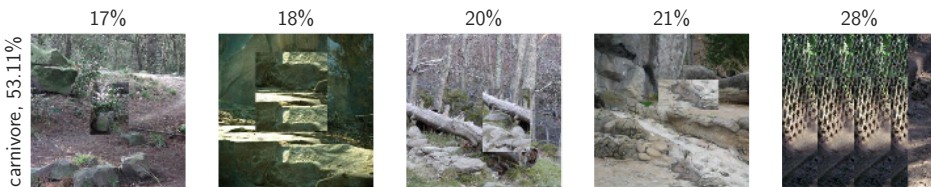

Figure 29: Most adversarial backgrounds—Carnivore.

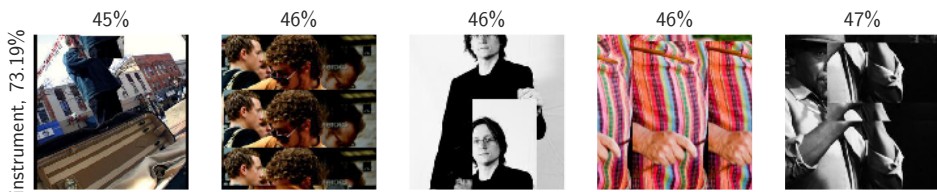

Figure 30: Most adversarial backgrounds—Instrument.

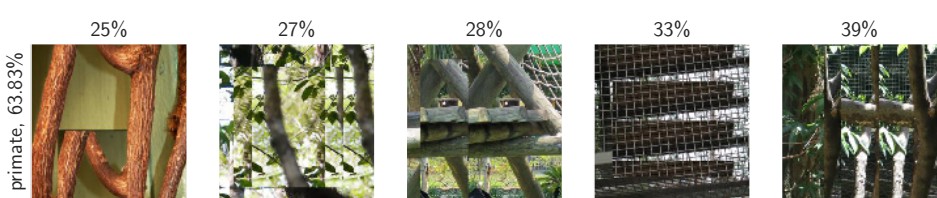

Figure 31: Most adversarial backgrounds—Primate.

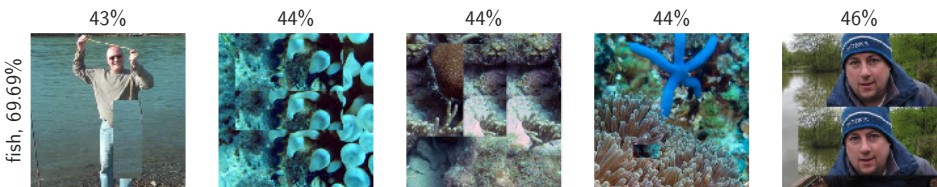

Figure 32: Most adversarial backgrounds—Fish.

## H    EXAMPLES OF FOOLING BACKGROUNDS IN UNMODIFIED IMAGES

We visualize examples of images where the background of the full original image actually fools models in Figure 33. For these images, models classify the foreground alone correctly, but they predict the same wrong class on the full image and the background. We denote these images as "BG Fools" in Table 3 and Figure 7. While this category is relatively rare (accounting for just 3% of the ORIGINAL-trained model's predictions), they reveal a subset of original images where background signal hurts classifier performance. Qualitatively, we observe that these images all have confusing or misleading backgrounds.

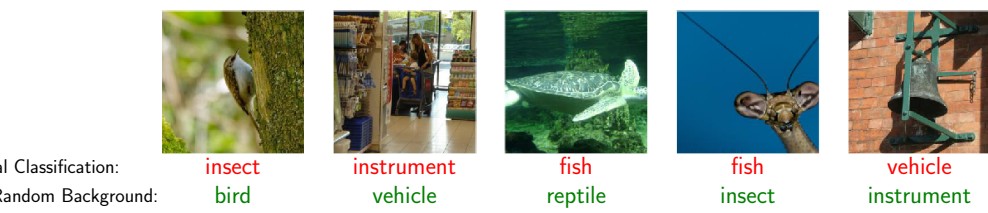

| | | | | | |
|---|---|---|---|---|---|
| Original Classification: | insect | instrument | fish | fish | vehicle |
| With Random Background: | bird | vehicle | reptile | insect | instrument |

Figure 33: Images that are incorrectly classified (as the class on the top row, which is the same class that their background alone from ONLY-BG-T is classified as), but are correctly classified (as the class on the bottom row) when the background is randomized. Note that these images have confusing backgrounds that could be associated with another class.

