# OpenReview forum: "Noise or Signal: The Role of Image Backgrounds in Object Recognition"
_ICLR.cc/2021/Conference — ICLR 2021 Poster_

### Official Review · AnonReviewer4 · 2020-10-26
**A very interesting work, but  possibly not original enough for an oral presentation. The work can attract a possible larger audience and get better feedback if presented as poster**

**Rating:** 8
**Confidence:** 5

**Review:**

I think this a very good contribution to ICLR given the topic and the quality of the submission (originality, contribution to the stare of the art, experimental evidence, etc) although the study might need to be supported in a more theoretical framework to make it worth of an oral presentation (I would recommend a poster or short presentation)

 Some of the strong points of the submission are summarized as follows:

1.	Studies in the interpretability of the results of deep learning models is a very important aspect, as well as the robustness of the obtained models in a variety of circumstances and under adversarial attacks.
2.	A sufficient introduction and motivations sections, but I would suggest introducing the state of the art at the beginning of the paper as it would help to get a better grasp of how the works builds upon previous work.
3.	The state of the art (despite the previous comment) contextualizes the subject matter in a succinct but comprehensive manner. Although there are certain aspects that could be improved, such as including a table outlining in a clearer manner the contributions of the authors in this context.
4.	The experimental design is good, showing a careful analysis to validate the proposal and several ablation studies to assess the validity of the authors' hypotheses
5.	The foundations for the method are presented in great detail in a formalized manner and provides sufficient elements (i.e. examples) to assess the validity of the proposed approach.


However, there are certain things that in my opinion could be improved:

1.	The authors make a very interesting contribution that leverages knowledge from several research areas and thus, sometimes the contributions with regards to the state of the art are difficult to follow. I would suggest making a table summarizing the main features of some previous works so the readers can better grasp the limitations of those previous works and understand better the improvements in each of the areas outlined in this research
2.	The organization of the paper is confusing, some effort should be given at creating a clearer layout that makes the paper easier to read and follow the flow of ideas.
3.	Future work could be further elaborated and discussion in specific domains (medical imaging, for instance) could be further discussed.
4.	The abstract mentions that the proposed work can be used as a blueprint for assessing the out of distribution performance of deep learning models, but this aspect is not sufficiently explored in my opinion.

---

> ### Author Response · Authors · 2020-11-18
> **Response to AnonReviewer4**
>
> Thank you for your time, your positive feedback, and your suggestions.
>
> =====
>
> As you suggested, we will revise our submission to discuss prior state of the art works more towards the beginning of the paper, and will also clearly highlight what we view as the main new contributions of the paper.
>
> =====
>
> Thank you for the suggestion. We will also include references to studies of background usage in medical imaging in our related works.
>
>
> =====
>
> Finally, regarding out-of-distribution performance, we view background dependence (and changing backgrounds) as probably the most basic example of distribution shift one can imagine. Thus, our study of backgrounds is just one particular case of distribution shift, but the questions we ask about backgrounds apply more generally to other forms of distribution shifts. We want to understand to what degree it is a problem in standard models, how it has changed over time, and how we can make models be more robust to it. Finally, we need a fine-grained benchmark for evaluating it, which is what we aimed to do for image backgrounds in this work.

---

### Official Review · AnonReviewer1 · 2020-10-27
**Official Blind Review #1**

**Rating:** 6
**Confidence:** 4

**Review:**

##########################################################################

Summary:

The paper studies the effect of background noise on image classification tasks for neural networks. The paper suggests the following based on empirical results from ImageNet classifiers.

1. It is possible to achieve reasonable accuracy by just using the background information.
2. Image classification models suffer from a decrease in accuracy if inference images have a different background.
3. Image classification models have higher accuracy tends to depend on the background image less.

The paper also introduces a toolkit for evaluating the ImageNet classifiers' dependence on background images.

##########################################################################

Pros:
1. The paper provides a detailed quantitative analysis of the effects of using different backgrounds (both training and testing). It constructs various synthetic test dataset that analyzes different scenarios.
2. The result sections (sections 3 and 4) are well structured and carefully study the impact of using different backgrounds.
3. The proposed toolkit can be used to evaluate the model's reliance on the background.

##########################################################################

Cons:
1. The key concern about the paper is the lack of novelty. While the synthetic dataset was constructed to study the effects of background in detail, the findings from the paper are not new.
2. The experiments do not study/relate how data augmentation techniques affect background reliance. The paper also mentions OOD techniques such as distributionally robust optimization (Sagawa et al, 2020), but does not study how these techniques affect background reliance. Moreover, I cannot find a discussion on which factors in training might force the model to less rely on image backgrounds (or robust to foreground images).
3. The dataset mainly used, IN-9, is also a small dataset that contains less than 50,000 train images. Moreover, the paper only considers the ImageNet type dataset. Some results may not hold for the other datasets. The authors do not address this in the manuscript.
4. There are some minor concerns about the experimental set-up used in the paper that I describe in the section below.
5. Writings can be significantly improved.

##########################################################################

Questions:
1. On page 5, it says, "MIXED-RAND models perform poorly on datasets with only backgrounds and no foregrounds." What is the insight from this experiment? Does this imply that the model might be learning shape features as it is doing better than random?
2. For images processed with GrabCut, wouldn't the model use shape-related features along with the background? If it is learning the shape features, can the positive correlation in section 4 mean that a stronger model might be learning more shape features?
3. What is each point in figure 8 represent? Is it using different architectures?
4. Appendix B.1, doesn't the change in the number of classes also result in a change in total dataset size?
5. What is the main insight of this paper when training neural networks?
6. What are the main contributions of the toolkit? How accurate is the segmentation?

Minor notes:
* On page 1, "standard models misclassify 88% of images ..." --> does this refer to 88% of test images?
* On page 1, I don't understand this sentence: "tend to simultaneously exploit background correlations more." Is there a stronger correlation between backgrounds for more accurate models?
* For figure 2, what is the test accuracy for a model trained on the original dataset (original/original)?
* In table 2, "on select test sets ..." --> selected.
* In figure 3, I don't understand what it means by "note that the gap decreases much more on the right side of the graph."
* It isn't easy to interpret table 3. It will be easier to understand if it contains some illustrative examples.

##########################################################################

I raised my score based on the author's response.

---

> ### Author Response · Authors · 2020-11-18
> **Response to AnonReviewer1 (1/3): Regarding Novelty**
>
> Thank you for your time and constructive feedback. We will first address the topic of novelty, then address further questions in the next comment.
>
> =====
>
> "The key concern about the paper is the lack of novelty. While the synthetic dataset was constructed to study the effects of background in detail, the findings from the paper are not new."
>
> As we state in our paper, we do not view the finding that models do use background info as the contribution of our work (nor a new finding altogether). We provide a number of relevant prior work citations in the introduction and related works about this topic.
>
> We do believe that the following are new contributions that appear first in our work, and welcome a discussion about which aspects are valuable. (As a note, many of the following comparisons are also in Appendix E.)
>
> Contributions we believe are novel in comparison to all prior works:
> 1) We study the relationship between the improvement of these newer architectures on ImageNet and their improvement on background-dependence in Section 4.
> 2) We analyze adversarial backgrounds and just how much models can use background signal. We find, interestingly, that models can be frequently fooled by adversarial backgrounds.
> 3) We analyze how different training methods (randomizing backgrounds during training, training with more data or more fine-grained classes) can decrease background dependence.
> 4) We analyze the BGs and FGs of individual images to show that backgrounds are actually required on certain images - this is Figure 7.
> 5) Our toolkit and code will become publicly available upon release of this work, so that others may also easily evaluate the background dependence of their own models. Our code is compatible with any ImageNet-trained model.
> 6) Our method for separating FGs and BGs can be achieved without human-annotated foreground segmentation. These human annotations can be expensive and do not exist for ImageNet.
>
> Next, we compare more explicitly with Zhu et. al specifically, the only other work we are aware of that also focuses on ImageNet:
> 1) Compared to Zhu et. al, we properly segment foregrounds using GrabCut (as opposed to using rectangular bounding boxes only, which may leave large portions of the background).
> 2) Compared to Zhu et. al, we combine FG and BG signal in various ways to gain a finer-grained understanding. For example, comparing MIXED-SAME with MIXED-RAND helps to isolate the effects of changing the background class. In contrast, Zhu et. al focuses on analyzing classification with only BG, which corresponds to just the ONLY-BG-B sub-dataset that we also include in our analysis.
> 3) Compared to Zhu et. al, which only studies AlexNet, we analyze a much larger set of more modern architectures.
>
> We would love to clarify this point better by including a more detailed discussion of the most relevant related works (e.g. Zhu et. al) earlier on in the paper, which Reviewer 4 also suggests.
>
> Would you find this helpful? Are there some other suggestions you might have?

---

> > ### Comment · AnonReviewer1 · 2020-11-20
> > **Response to authors**
> >
> > Thank you for the detailed reply. I carefully read your response and other reviewers' feedback.
> >
> > Regarding novelty:
> >
> > Thank you for clarifying the novelty of this paper. I still think some contributions (e.g. 2, 3) are not new observations, but I acknowledge that the paper quantitively studies these effects, and these results will help the research community. I also recognize that the toolkit for analyzing the background would also be useful to the research community. As suggested, a more detailed discussion of the most relevant related works earlier in the paper would be helpful.
> >
> > "Yes, we agree that one conclusion could be that models are processing shape features better. In particular, better models also perform better on NO-FG, which has foreground shape but no texture."
> >
> > It would be helpful to discuss (clarify) these points in the manuscript, as the background dependence is not the only conclusion from these experiments.
> >
> > Other points:
> >
> > Thank you for clarifying all my questions. I agree with the author's responses. I suggest authors to (1) summarize the main features of the previous works and highlight the contributions (difference) of this work, (2) improve the clarify the writings and presented results in the next revision. I raised my score accordingly.

---

> > > ### Author Response · Authors · 2020-11-20
> > > **Response to AnonReviewer1**
> > >
> > > Thank you again for your time, suggestions, and for carefully reading and considering our response and others' feedback.
> > >
> > > We have updated our submission with the suggestions you recommended. We specifically discuss the most relevant related work in the introduction of our work, and discuss all other related works and what we view as our contributions in comparison to those works in greater detail in the Related Works section.
> > >
> > > We have also worked to clarify confusing sentences that you pointed out, and added helpful figures such as illustrative examples for Table 3.
> > >
> > > Finally, thank you for the suggestion regarding the topic of improved use of shape features (as Figure 8 implies). We discuss it in our revision.

---

> ### Author Response · Authors · 2020-11-18
> **Response to AnonReviewer1 (2/3): Answering Specific Questions**
>
> We now address specific questions and comments that you brought up. We are happy to discuss and clarify further if you would find it helpful.
>
> =====
>
> "Moreover, the paper only considers the ImageNet type dataset. Some results may not hold for the other datasets. The authors do not address this in the manuscript."
>
> You are right that we specifically choose to focus on ImageNet. While other datasets are analyzed in other works, we find it important to focus on the biggest and most widely used context (ImageNet dataset) and to provide the most comprehensive (as well as extensible) analysis. Furthermore, many pre-trained models for it are already in use. The only prior work that has a similar focus on ImageNet is the work of Zhu et. al., and, as discussed in the Related Works section (and in the previous response), there are a number of new insights that our work contributes compared to it.
>
>
> =====
>
> "The experiments do not study/relate how data augmentation techniques affect background reliance. The paper also mentions OOD techniques such as distributionally robust optimization (Sagawa et al, 2020), but does not study how these techniques affect background reliance. Moreover, I cannot find a discussion on which factors in training might force the model to less rely on image backgrounds (or robust to foreground images)."
>
> We discuss various factors in training that affect background robustness. In Section 3, we discuss how training with (1) randomized backgrounds (MIXED-RAND) improves robustness to changing backgrounds and (2) larger training dataset sizes decreases background dependence. We then discuss in Section 4 how newer architectures that do better on ImageNet are also more background-robust.
>
> Without our toolkit, it would be hard to evaluate the effect of different training procedures on background-robustness. Now, one can use our toolkit to do that and study other questions related to background-robustness, in particular, and background dependence, in general. Thus, studying the effects of different training techniques on background dependence is much more feasible using the toolkit we introduce.
>
>
> =====
>
> "The dataset mainly used, IN-9, is also a small dataset that contains less than 50,000 train images."
>
> Many of the models we evaluate are either pre-trained ImageNet models (trained on 1.2 million images from ImageNet) or models trained on IN-9L, which is a 180k image dataset consisting of all ImageNet classes corresponding to our test set. For example, all of Table 2 and all of Section 4 are focused on evaluating such models.
>
> The reason we need to make a smaller IN-9 train set is so that we can train on the synthetic variations of IN-9 as well. Creating these synthetic variations requires bounding boxes, and since not all images in ImageNet have bounding boxes, these training sets must be smaller. Then, for example, when comparing models trained on ORIGINAL and MIXED-RAND (e.g. in Figure 5), we ensure that any differences we observe are not a result of dataset size, which is held constant, but instead are a result of the differences in backgrounds between ORIGINAL and MIXED-RAND.
>
> =====
>
> "On page 5, it says, "MIXED-RAND models perform poorly on datasets with only backgrounds and no foregrounds." What is the insight from this experiment? Does this imply that the model might be learning shape features as it is doing better than random?"
>
> That means it is not effectively using background signal (as compared to all other models, e.g. the model trained on regular images, ORIGINAL). This is not surprising, and could be a desired effect if the goal is background invariance. In Appendix D, we discuss why MIXED-RAND does slightly better than random; it may still use some information -- e.g. there may be detectable artifacts such as the “size” of the foreground image that exist in the ONLY-BG-T test set that the MIXED-RAND model also learns to use (e.g. if it can tell roughly the size of the foreground, it might use that). However, the key thing is to compare MIXED-RAND to other models, and MIXED-RAND uses background the least, as one would expect.
>
>
> =====
>
> "For images processed with GrabCut, wouldn't the model use shape-related features along with the background? If it is learning the shape features, can the positive correlation in section 4 mean that a stronger model might be learning more shape features?"
>
> Yes, we agree that one conclusion could be that models are processing shape features better. In particular, better models also perform better on NO-FG, which has foreground shape but no texture.
>
> =====
>
> "What is each point in figure 8 represent? Is it using different architectures?"
>
> Each “x” is a specific model architecture, evaluated on a specific dataset. So each vertical line (consisting of 5 points) is 1 architecture, evaluated on 5 different test sets (e.g. see the vertical line corresponding to MobileNet).

---

> ### Author Response · Authors · 2020-11-18
> **Response to AnonReviewer1 (3/3): Answering Specific Questions**
>
> "Appendix B.1, doesn't the change in the number of classes also result in a change in total dataset size?"
>
> It does not. We change the number of classes by grouping together ImageNet classes into higher-level classes (e.g. labeling all 16 dog breeds as the “dog” class), but we keep the set of images the same. So when we have 144 classes (n=1), each class has about 1.3k images. When we have 9 classes (n=16), each class has about 20k images.
>
>
> =====
>
> "What is the main insight of this paper when training neural networks?"
>
> Some insights we find interesting are that better ImageNet models are more background-robust (and thus, progress on ImageNet accuracy has also corresponded to progress in background-robustness), that training with larger dataset sizes can somewhat improve background robustness, and training with randomized backgrounds (e.g. MIXED-RAND) can improve background robustness greatly.
>
>
> =====
>
> "What are the main contributions of the toolkit? How accurate is the segmentation?"
>
> We disentangle ImageNet FGs and BGs in a fully automated way using both the bounding boxes and the GrabCut algorithm. We then create different combinations of FGs and BGs so that background dependence can be investigated in a fine-grained manner. The segmentation is fairly accurate -- we provide samples of random images displaying the segmentation and all datasets variations in Appendix F (these are like Figure 1, but for different classes).
>
> We acknowledge that the segmentation is not perfect, which is why we focus so much on the BG-GAP, the difference between MIXED-SAME and MIXED-RAND, rather than the gap between ORIGINAL and MIXED-RAND. This is because any imperfections from the segmentation will exist in both MIXED-SAME and MIXED-RAND, and so the only difference between the two is the backgrounds being used (which is exactly what we seek to measure).
>
> =====
>
>
> Thank you for your minor notes as well; we took your suggestions and also respond to your questions here.
>
> =====
>
> "On page 1, "standard models misclassify 88% of images ..." --> does this refer to 88% of test images?"
>
> Yes. 88% of test images are misclassified when their FGs are combined with adversarial BGs.
>
>
> =====
>
> "On page 1, I don't understand this sentence: "tend to simultaneously exploit background correlations more." Is there a stronger correlation between backgrounds for more accurate models?"
>
> Thank you for pointing out the ambiguity in this sentence -- we will clarify the sentence to state that more accurate models can predict with higher accuracy using just the backgrounds of images. The background correlations (e.g. the correlation between flower backgrounds and the insect class) are a property of the images; they exist independent of the model, and more accurate models can exploit/use these correlations to achieve higher accuracy when presented with images that only have backgrounds.
>
> =====
>
> "In figure 3, I don't understand what it means by "note that the gap decreases much more on the right side of the graph.""
>
> The gap between MIXED-RAND and MIXED-SAME becomes much lower when the training dataset set size is largest (on the right side of the graph). We will clarify this in the revised version.
>
> =====
>
> "It isn't easy to interpret table 3. It will be easier to understand if it contains some illustrative examples."
>
> Thank you for the suggestion; we will include illustrative examples in the revision.

---

### Official Review · AnonReviewer3 · 2020-10-28
**Interesting empirical study quantifying Imagenet-trained models' reliance on background on Imagenet**

**Rating:** 5
**Confidence:** 3

**Review:**


The submission performs similar foreground-background analysis for object recognition as in [1], but with more modern networks in mind. As such, the main takeaways indicate that this phenomenon still exists - networks today continue to suffer from background bias as they did four years ago with AlexNet, although maybe to a lesser extent. This submission curates more careful evaluation setups by using segmentation of foreground objects, tiled backgrounds to create multiple datasets that serve to illustrate the trends in a more disambiguated way.

Is there some way to quantify the overall diversity of adversarial backgrounds? For example, is it possible that owing to strong correlations of a few objects with easily-learned backgrounds, these few backgrounds always cause misclassification for other objects? Could there be a way to detect such backgrounds?

The Appendix says "The ORIGINAL-trained model performs similarly on NO-FG and ONLY-BG-B, indicating that it does not use object shape effectively”  but there seems to be a 10% gap in Table 5, indicating that the shape mask is fairly useful. The IN-9L numbers seem 21% up instead of 13%. Am I misreading this table?

Re. "Indeed, the ONLY-BG trend observed in Figure 8 suggests that…”, could an additional possibility be that around 20% of  classes are fully correlated with their backgrounds? I.e. how can we know how much of the findings are to do with model "failure", and not dataset quirks?

In summary,

(+) While it is not particularly surprising to learn that backgrounds can be misleading even with the correct foreground or that there exists vulnerability to adversarially picked backgrounds, given the evidence we have of background biases already, it can be useful to have a quantification of the “BG-gap” for a range of modern models.

(-) The takeaways are mostly already recognised from the many works that have pointed out reliance of object recognition models on backgrounds. The experiments provide a quantified view of how much modern networks trained on the particular datasets rely on backgrounds, but it is unclear how widely applicable this information is, given that this only analyses a specific dataset. The curated datasets might be useful for benchmarking progress; however, if one were to set up the goal of providing such a testset, then perhaps it might be more appropriate to curate an entire testset of adversarial backgrounds alone (rather than mixed-rand) across a range of modern networks and for all of ImageNet, which, along with the usual test set would provide a background-robustness sanity check (with the caveat from the authors that backgrounds may actually be informative when the foreground is confusing).


[1] Object recognition with and without objects, Zhu et al. 2017

Post Rebuttal:
I appreciate the authors' responses. The "novelty" over Zhu et al. was never under question in my review, I was mostly confused about how to weigh the significance of the findings, how useful it is to know some numbers for the version of the dataset created by the authors (which is not really the original Imagenet classification task), and if the submission actually does "pinpoint" what the problems are, how and when they manifest, to what extent the dataset is responsible vs. the training choices. Having read the other reviews, responses, looked at the updates, I'm still unsure --  if there were something in this paper that was new or surprising and not more or less already known from existing works (perhaps not precise numbers, but then the paper is essentially using a synthetic, modified Imagenet anyway), I'd be more enthusiastic about pushing up the rating. But as of now, I'm retaining my initial rating.

---

> ### Author Response · Authors · 2020-11-18
> **Response to AnonReviewer3 (1/2): Regarding Novelty**
>
> Thank you for your time and constructive feedback. We will first address the topic of novelty, then address your specific questions in the next comment.
>
> =====
>
> "(-) The takeaways are mostly already recognised from the many works that have pointed out reliance of object recognition models on backgrounds. The experiments provide a quantified view of how much modern networks trained on the particular datasets rely on backgrounds, but it is unclear how widely applicable this information is, given that this only analyses a specific dataset."
>
> We believe that, given the growing interest in out of distribution classification and understanding model biases, it’s important to have a toolkit for pinpointing the exact challenges and evaluating our progress on these fronts. Our focus on ImageNet is motivated by the fact that it is the most widely used dataset for academic research, and many pre-trained models for it are already in use. The only prior work that has a similar focus on ImageNet is the work of Zhu et. al., and, as discussed in the Related Works section (and in the next paragraph), there are a number of new insights that our work contributes compared to it.
>
> We do not claim to be the first to point out that models can rely on backgrounds--many works that we cite already study this. We do believe that the following are new contributions that appear first in our work, and welcome a discussion about which aspects are valuable. (As a note, many of the following comparisons are also in Appendix E.)
>
> Specific comparison to Zhu et. al:
> 1) Compared to Zhu et. al, we properly segment foregrounds using GrabCut (as opposed to using rectangular bounding boxes only, which may leave large portions of the background).
> 2) Compared to Zhu et. al, we combine FG and BG signal in various ways to gain a finer-grained understanding. For example, comparing MIXED-SAME with MIXED-RAND helps to isolate the effects of changing the background class. In contrast, Zhu et. al focuses on analyzing classification with only BG, which corresponds to just the ONLY-BG-B sub-dataset that we also include in our analysis.
> 3) Compared to Zhu et. al, which only studies AlexNet, we analyze a much larger set of more modern architectures.
>
> Contributions we believe are novel in comparison to all prior works (including Zhu et. al):
> 1) We study the relationship between the improvement of these newer architectures on ImageNet and their improvement on background-dependence in Section 4.
> 2) We analyze adversarial backgrounds and just how much models can use background signal. We find, interestingly, that models can be frequently fooled by adversarial backgrounds.
> 3) We analyze how different training methods (randomizing backgrounds during training, training with more data or more fine-grained classes) can decrease background dependence.
> 4) We analyze the BGs and FGs of individual images to show that backgrounds are actually required on certain images - this is Figure 7.
> 5) Our toolkit and code will become publicly available upon release of this work, so that others may also easily evaluate the background dependence of their own models. Our code is compatible with any ImageNet-trained model.
> 6) Our method for separating FGs and BGs can be achieved without human-annotated foreground segmentation. These human annotations can be expensive and do not exist for ImageNet.

---

> ### Author Response · Authors · 2020-11-18
> **Response to AnonReviewer 3 (2/2): Answering Specific Questions**
>
> We will now answer the specific questions that you asked, and are happy to clarify further if it is helpful for you.
>
>
> =====
>
> "Is there some way to quantify the overall diversity of adversarial backgrounds? For example, is it possible that owing to strong correlations of a few objects with easily-learned backgrounds, these few backgrounds always cause misclassification for other objects? Could there be a way to detect such backgrounds?"
>
> In Appendix G (Figure 24), we present a distribution of how adversarial all insect backgrounds (BGs) are. In particular, we observe both (1) many BGs cause models to misclassify FG+BG combinations as the BG class (insect) on at least 10% of all FG objects (non-insect objects). For the insect class, more than ⅓ of BGs have this property; and (2) BGs have a wide range of adversarial attack success rates--some are extremely effective at causing misclassification as the BG class, while others are ineffective. We agree that the adversarial effectiveness of some BGs is likely due to easily-learnable BGs being highly correlated with the class. For example, in Figure 4, we displayed 5 BGs containing flowers, which are highly correlated with the insect class and cause misclassification for other objects.
> Our methodology for detecting these adversarial BGs is simple - we combine every FG with every BG in our dataset, and record the classification output of models for every FG/BG combination. We detect the most adversarial BGs by checking which BGs fool the greatest number of FG objects. Appendix G has examples of these “most adversarial” backgrounds for every class. If there is some information on this topic you would find helpful that is not in Appendix G or in the main body of the paper, or if we misunderstood what you were asking, we are happy to include more information or clarify further.
>
>
> =====
>
> "The Appendix says "The ORIGINAL-trained model performs similarly on NO-FG and ONLY-BG-B, indicating that it does not use object shape effectively” but there seems to be a 10% gap in Table 5, indicating that the shape mask is fairly useful. The IN-9L numbers seem 21% up instead of 13%. Am I misreading this table?"
>
> Thank you for correcting us - we mistakenly copied the numbers for NO-FG vs. ONLY-BG-T (as opposed to ONLY-BG-B). After correcting this, it is now a 21% vs. 10% gap, which still gives the same conclusion that the IN-9L-trained model, which uses more training data, improves performance when the object shape is involved. Both models perform fairly similarly when shown backgrounds without object shape (33% vs. 34% for ORIGINAL-trained vs. for IN-9L-trained), while the IN-9L-trained model does relatively much better once object shape is included (42% vs. 56%). We will correct this in the revision to state that the ORIGINAL-trained model uses object shape, but the IN-9L-trained model uses object shape more effectively.
>
>
> =====
>
> "Re. "Indeed, the ONLY-BG trend observed in Figure 8 suggests that…”, could an additional possibility be that around 20% of classes are fully correlated with their backgrounds? I.e. how can we know how much of the findings are to do with model "failure", and not dataset quirks?"
>
> We fully agree that this is not a model failure mode per se, and it could just be that some images require using the background to correctly classify (ImageNet may have quirks that make backgrounds necessary in some cases). Still, we believe it is valuable to pinpoint such dependence. We will clarify this paragraph in the revision to clearly acknowledge that this is not a model failure mode per se.
>
>
> =====
>
> "The curated datasets might be useful for benchmarking progress; however, if one were to set up the goal of providing such a testset, then perhaps it might be more appropriate to curate an entire testset of adversarial backgrounds alone (rather than mixed-rand) across a range of modern networks and for all of ImageNet, which, along with the usual test set would provide a background-robustness sanity check (with the caveat from the authors that backgrounds may actually be informative when the foreground is confusing)."
>
> As for testing on adversarial backgrounds (in addition to Mixed-Rand), the full code repo that we will release with the paper does indeed provide code to evaluate your own models against the entire set of adversarial backgrounds.
> Note: The anonymized code repo that we linked to at submission time does not contain the code for adversarial background evaluation, but we are happy to share the script for doing so that will come with the full code release. We intend for this code to be viewed as a background-robustness sanity check that anyone can use to better understand their models’ behavior.

---

> > ### Comment · AnonReviewer3 · 2020-11-19
> > **adversarial backgrounds**
> >
> > I think I was more interested in getting a sense for how certain types of backgrounds can cause problems with generalisation for ImageNet. So rather than the specific background pictures that are shown to be most adversarial for IN-9L, I was wondering if there could be a way to show, for every ImageNet category, that there are common types of backgrounds that pose problems (for example, x% of insect pictures have flower-backgrounds). I realise this is a hard thing to quantify, since backgrounds are not labelled. Appendix G goes some way towards such analysis, but it feels a bit limited (hence possibly anecdotal).
> >
> > How would the authors say the datasets curated in "Natural Adversarial Examples" (Hendrycks et al.) compare to the proposed toolkit, in terms of assessing robustness (one would typically want to be robust to more factors than are contained in the background alone)? Do the authors believe backgrounds are more of a problem than other distracting factors (pose, colour, texture, etc)?

---

> > > ### Author Response · Authors · 2020-11-20
> > > **Response regarding adversarial backgrounds and Natural Adversarial Examples**
> > >
> > > On understanding adversarial backgrounds:
> > >
> > > Thank you for the clarification---we agree that this is difficult to quantify since backgrounds are not labeled. Thus, we feel that the methodology used to generate Appendix G (combine each background with each foreground, and inspect the backgrounds that have the highest attack success rate) is a good way to get a sense of what types of backgrounds commonly pose problems for each class.
> > >
> > > =====
> > >
> > > On "Natural Adversarial Examples" (Hendrycks et al.)
> > >
> > > We agree with the reviewer in that "Natural Adversarial Examples" (NAE) consider many factors of variation, whereas we focus only on backgrounds. However, this is actually by design: while the datasets in the NAE paper provide convincing evidence that modern deep learning systems are non-robust, the goal of our dataset is to begin to understand why models are so non-robust. Doing so requires considering the factors of variation separately and understanding each one’s effect on machine learning classifiers, which is a distinct (but we believe, still worthwhile) objective from that of the NAE paper.
> > >
> > > We do not compare backgrounds with other distracting factors, but we believe image backgrounds are the most basic and natural example of a correlation between images and labels that models may use in a different way from humans.

---

### Official Review · AnonReviewer2 · 2020-10-29
**A comprehensive study on the role of background in image classification**

**Rating:** 7
**Confidence:** 3

**Review:**

The authors presented a comprehensive study on the role of background in image classification. They designed a new set of data and a lot of experiments to find answers to the following questions: (1) How much decrease in classification accuracy if the background signal is removed? (2) Can a model successfully classify an image solely based on its background? (3) Will an image be misclassified if the image's background is replaced by a different background? (4) With the advance of the model architecture, are the more advanced models like ResNet more robust to background effect?

The paper is well written, and the figures and tables are clearly presented. The newly created dataset ImageNet-9, that contains background- and foreground-free images, are publicly available.

Comments:
- I am not sure if I understand the purpose for presenting the ONLY-BG-T results in Figure 7. In my opinion, by comparing the BG-Required numbers in MIXED-RAND and ORIGINAL models, it is already clear enough to demonstrate that the background is a necessary component for many images to obtain correct classification.
- As shown in the section of related work, similar topics have been studied before. One of the main contributions of this paper is the newly created dataset. It can be generally useful for ML research in robustness and out-of-distribution detection.

---

> ### Author Response · Authors · 2020-11-18
> **Author Response for AnonReviewer2**
>
> Thank you for the positive feedback!
>
> You are correct that presenting results comparing the BG-Required numbers for ORIGINAL vs. MIXED-RAND already conveys the main point that backgrounds are more necessary for models trained on ORIGINAL. We include a ONLY-BG-T-trained model in the same plot for completeness, as the 5 categories of images (BG Irrelevant, BG Required, BG Fools, BG+FG Required, BG+FG Fools) are determined by test performance on the 3 datasets of ORIGINAL, MIXED-RAND, and ONLY-BG-T.

---

### Author Response · Authors · 2020-11-20
**Summary of Changes in Revision**

We again thank all reviewers for their valuable feedback.

We updated our submission with the following key changes:

- Discussion of the most relevant related works (e.g. Zhu et. al 2017) in the introductory section, and clarification of what we view as our key contributions in relation to prior works
- More detailed comparisons to prior works in the Related Works section
- Clarification of confusing sentences and explanations
- Addition of illustrative examples when appropriate
- Minor corrections and typos

Furthermore, we have now uploaded the full submission (main paper + the appendix) as a single pdf file.

---

### Decision · Program_Chairs · 2021-01-07
**Final Decision**

**Decision:**

Accept (Poster)

**Comment:**

The paper investigates the tendency of image recognition models to depend on image backgrounds, and propose a suite of datasets to study this phenomenon.

All the reviewers agree that the paper investigates an important problem, is well-written and contains several interesting insights that should be of interest to the community. I recommend acceptance.